# Soft Equivariance Regularization for Invariant Self-Supervised Learning

**Joohyung Lee**[1]    **Changhun Kim**[1]    **Hyunsu Kim**[2]    **Kwanhyung Lee**[1,3]    **Juho Lee**[3]

[1]AITRICS
[2]Voronoi Inc.
[3]Korea Advanced Institute of Science and Technology (KAIST)
{chris,changhun.kim,destin}@aitrics.com
hk@voronoi.io, juholee@kaist.ac.kr

## Abstract

Self-supervised learning (SSL) typically learns representations invariant to semantic-preserving augmentations (e.g., random crops and photometric jitter). While effective for recognition, enforcing strong invariance can suppress transformation-dependent structure that is useful for robustness to geometric perturbations and spatially sensitive transfer. A growing body of work, therefore, augments invariance-based SSL with equivariance objectives, but these objectives are often imposed on the same *final* (typically spatially-collapsed) representation. We empirically observe a trade-off in this coupled setting: pushing equivariance regularization toward deeper layers improves equivariance scores but degrades ImageNet-1k linear evaluation, motivating a layer-decoupled design. Motivated by this trade-off, we propose **Soft Equivariance Regularization (SER)**, a plug-in regularizer that decouples where invariance and equivariance are enforced: we keep the base SSL objective unchanged on the final embedding, while softly encouraging equivariance on an *intermediate spatial token map* via analytically specified group actions $\rho_g$ (e.g., $90°$ rotations, flips, and scaling) applied directly in feature space. SER learns/predicts no per-sample transformation codes/labels, requires no auxiliary transformation-prediction head, and adds only **1.008×** training FLOPs. On ImageNet-1k ViT-S/16 pretraining, SER improves MoCo-v3 by **+0.84** Top-1 in linear evaluation under a strictly matched 2-view setting and consistently improves DINO and Barlow Twins; under matched view counts, SER achieves the best ImageNet-1k linear-eval Top-1 among the compared invariance+equivariance add-ons. SER further improves ImageNet-C/P by **+1.11/+1.22** Top-1 and frozen-backbone COCO detection by **+1.7** mAP. Finally, applying the same **layer-decoupling** recipe to existing invariance+equivariance baselines (e.g., EquiMod and AugSelf) improves their accuracy, suggesting layer decoupling as a general design principle for combining invariance and equivariance. Code is available at `https://github.com/aitrics-chris/SER`.

## 1 Introduction

Self-supervised learning (SSL) is a central paradigm for visual representation learning (Chen et al., 2020; Caron et al., 2021; Wang et al., 2023), enabling strong features to be learned from large-scale unlabeled data. Many successful SSL methods follow a simple principle: learn representations that are *invariant* to a distribution of semantic-preserving augmentations, such as random crops and photometric jitter. This invariance is highly effective for recognition, but it can also suppress transformation-dependent structure (e.g., orientation, reflection, or scale cues) useful for geometric robustness and spatially sensitive transfer. A complementary principle is *equivariance*: rather than discarding transformation information, an equivariant representation changes in a predictable and structured way when the input is transformed (Dangovski et al., 2021; Marchetti et al., 2023).

Prior work that incorporates equivariance into SSL can be broadly grouped into two families (Yu et al., 2024). *Implicit* approaches encourage transformation awareness through auxiliary objectives (e.g., predicting or contrasting transformations) (Dangovski et al., 2021; Lee et al., 2021). *Explicit*

approaches aim to model how features transform in latent space, often introducing auxiliary transformation/action modules or learning/predicting per-sample transformation codes/labels as training targets (Devillers & Lefort, 2023; Park et al., 2022; Garrido et al., 2023; Yu et al., 2024). At ImageNet scale, equivariant SSL has also been explored in predictive/world-model settings (Garrido et al., 2024). In this work, we study a complementary problem: *how to incorporate an explicit equivariance regularizer on top of strong invariance-based SSL backbones (MoCo-v3, DINO, Barlow Twins) for ViTs, without changing the architecture or the baseline invariance objective.*

A common design choice in prior invariant+equivariant SSL methods is to impose both objectives on the *final* representation. However, the final representation is typically spatially collapsed, and thus poorly aligned with spatial group actions. More importantly, we empirically observe a trade-off when equivariance regularization is pushed toward the output: equivariance scores increase, but ImageNet-1k linear-evaluation accuracy consistently decreases (Figure 3 and table 4). This motivates a decoupled design in which invariance and equivariance are encouraged *at different layers*.

We propose **Soft Equivariance Regularization (SER)**, a simple yet scalable regularizer that **decouples** where invariance and equivariance are enforced. SER trains the final embedding with an invariant baseline SSL objective, while softly regularizing *spatially structured intermediate representations* toward equivariance via analytically known feature-space group actions $\rho_g$. Since some common SSL augmentations (notably random cropping) are non-invertible and do not form a group, SER uses a simple batch-partition strategy: the full augmentation pipeline contributes to the invariance objective, while equivariance is enforced only for an invertible geometric subset.

**Transformation labels.**  Following the terminology of STL (Yu et al., 2024), by "no transformation labels" we mean that SER does not *learn or predict* per-sample transformation labels/codes as additional supervision. SER still assumes the standard (known) augmentation pipeline and uses the resulting relative transform $g = g_2 g_1^{-1}$, but does not introduce a transformation-prediction head.

We evaluate SER on ImageNet-1k pretraining of ViT-S/16 and a suite of robustness and transfer benchmarks. Because the number of views can significantly affect SSL performance, we emphasize *matched-view* comparisons and report results under strictly matched settings. Across strong invariance-based SSL methods (MoCo-v3, DINO, Barlow Twins), SER consistently improves ImageNet-scale classification and robustness (e.g., ImageNet-C/P), and yields larger gains on spatially sensitive transfer such as frozen-backbone COCO detection. Furthermore, we show that the same **layer-decoupling** modification can improve existing invariance+equivariance baselines, suggesting a general recipe for combining invariance and equivariance in practice.

**Contributions.**  Our contribution is fourfold:

- **Empirical trade-off at the final layer.** We show that imposing invariance and equivariance on the *same final representation* can be suboptimal: pushing equivariance regularization toward later layers increases equivariance scores but reduces ImageNet-1k linear-evaluation accuracy (Figure 3 and table 4).
- **Layer-decoupled soft equivariance regularization.** We propose SER, which promotes equivariance on an intermediate spatial representation while leaving the baseline SSL loss unchanged on the final embedding.
- **Analytic feature-space group actions without extra modules.** SER applies analytically specified actions $\rho_g$ to intermediate feature maps, avoiding auxiliary transformation/action modules and avoiding learned/predicted per-sample transformation codes/labels.
- **Layer decoupling upgrades prior inv+equiv methods.** Beyond our loss design, we show that moving the equivariance objective of prior invariant+equivariant methods (e.g., EquiMod, AugSelf) from the final layer to an intermediate layer improves their accuracy as well, indicating that layer decoupling is broadly beneficial.

## 2 BACKGROUNDS

### 2.1 SELF-SUPERVISED LEARNING

Self-supervised learning (SSL) leverages supervisory signals derived from the data itself, reducing reliance on human-annotated labels. Early SSL methods used pretext tasks such as rotation predic-

tion (Gidaris et al., 2018) or solving jigsaw puzzles (Noroozi & Favaro, 2016). Modern SSL in vision predominantly learns representations by enforcing *invariance* across augmented views of the same image, often via instance discrimination or redundancy-reduction objectives (Chen et al., 2020; He et al., 2020; Grill et al., 2020; Zbontar et al., 2021; Bardes et al., 2022). Concretely, a training sample is transformed into multiple views by independently sampling augmentations from a predefined distribution (e.g., crops and photometric jitter), and the model is optimized so that representations of these views agree.

Different SSL frameworks instantiate this invariance principle with different objectives. Contrastive approaches such as SimCLR and MoCo (including MoCo-v3) use noise-contrastive estimation losses (Chen et al., 2020; He et al., 2020; Chen & He, 2021), while non-contrastive methods such as BYOL and SimSiam minimize a similarity loss between predicted and target embeddings (Grill et al., 2020; Chen & He, 2021). Redundancy-reduction methods such as Barlow Twins and VICReg enforce agreement while discouraging collapse through covariance-based regularization (Zbontar et al., 2021; Bardes et al., 2022). Many frameworks use two global views by default, while increasing the number of views (e.g., multi-crop training) can improve performance at the cost of additional compute and memory (Caron et al., 2020; 2021).

SER is designed to be complementary to these invariance-based objectives: we keep the base SSL loss unchanged on the final embedding, and add an auxiliary *equivariance* regularizer applied to an intermediate spatial representation (Section 3.3).

## 2.2 EQUIVARIANT REPRESENTATION LEARNING

Invariant SSL encourages representations to ignore augmentation-induced variability, which can be beneficial for recognition but may suppress transformation-dependent structure. Equivariant representation learning instead aims to encode how representations should change under transformations in a structured way, complementing invariance (Dangovski et al., 2021; Marchetti et al., 2023). Recent approaches that incorporate equivariance into SSL can be broadly grouped into *implicit* and *explicit* families (Yu et al., 2024).

**Implicit methods.** Implicit approaches encourage transformation awareness via auxiliary objectives, for example by predicting transformation parameters/labels or contrasting transformations across views (Dangovski et al., 2021; Lee et al., 2021). While effective for certain transformations, such objectives can be challenging to scale to complex compositions or to disentangle multiple simultaneously applied augmentations.

**Explicit methods.** Explicit approaches attempt to model how features transform in latent space, often by introducing auxiliary transformation/action modules or by learning/predicting per-sample transformation codes/labels as training targets (Devillers & Lefort, 2023; Garrido et al., 2023; Yu et al., 2024). A common practical design is to impose equivariance constraints on a *spatially collapsed* representation (e.g., pooled CNN features or a ViT `[CLS]` token), which can limit sensitivity to fine-grained geometric changes and makes spatial group actions less directly applicable.

**SER.** In this work, we focus on a simple and scalable alternative for ViTs: we impose a *soft* equivariance regularizer on an intermediate, spatially structured representation (a token map), while keeping the final embedding optimized purely by the original invariance-based SSL objective. This decoupling avoids applying equivariance to a collapsed representation and avoids learning/predicting transformation codes/labels or training auxiliary transformation-prediction heads.

## 2.3 SYMMETRY, GROUPS, AND EQUIVARIANCE

Symmetry refers to a transformation that preserves relevant structure of an object (Bronstein et al., 2021). For example, rotating a perfect circle around its center does not alter its appearance. The set of such transformations can form a *group* $\mathcal{G}$ under a binary operation (here, composition), satisfying closure, associativity, an identity element, and inverses.

To formalize transformation behavior, we consider *group representations*. A (linear) representation is a homomorphism $\rho : \mathcal{G} \to \mathrm{GL}(n)$; we denote $\rho(g)$ by $\rho_g$. A function $f : \mathcal{X} \to \mathcal{Y}$ is $\mathcal{G}$-*equivariant* if, for all $g \in \mathcal{G}$ and $x \in \mathcal{X}$,

$$f\big(\rho_g(x)\big) = \rho_g\big(f(x)\big). \tag{1}$$

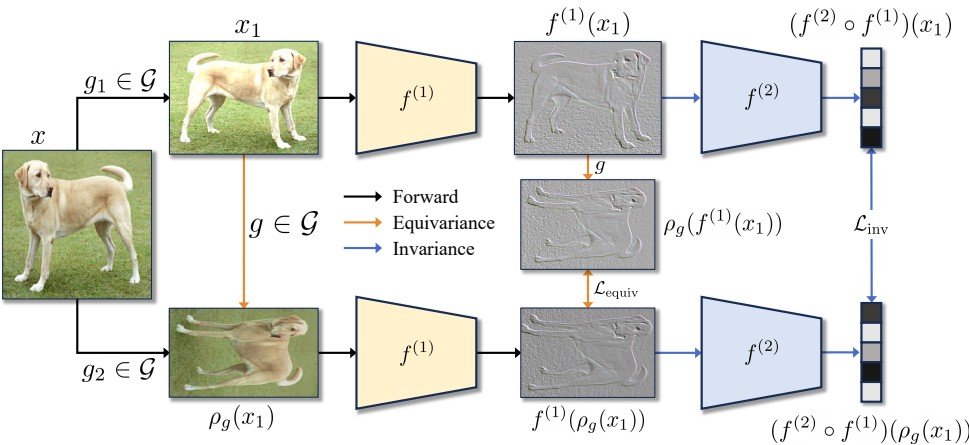

Figure 1: Overview of SER. For each image in $b_2$, we sample two views from the equivariant-view policy $\mathcal{T}_{\text{eq}}$. We decompose each sampled transform into a geometric component $g \in \mathcal{G}$ and a photometric component (e.g., color jitter), and denote by $g_1, g_2 \in \mathcal{G}$ the geometric parts of the two views. We use the relative transform $g = g_2 g_1^{-1}$ to align intermediate token maps in feature space. For clarity, the independently sampled photometric components in $\mathcal{T}_{\text{eq}}$ are omitted in the diagram.

For simplicity, we overload $\rho_g$ to denote the induced actions on both the input and output spaces; in practice these actions may differ. Note that invariance is a special case of equivariance where $\rho_g$ is the identity map for all $g$.

In practice, many models are only approximately equivariant. *Hard* equivariance is enforced by architecture design (e.g., group-equivariant CNNs (Cohen & Welling, 2016)), whereas *soft* equivariance is encouraged by a regularization term weighted by a coefficient (Finzi et al., 2021; Kim et al., 2023). CNNs provide a canonical example: convolutional layers are (approximately) equivariant to translations, up to discretization and boundary effects. Motivated by such symmetries, many architectures encode broader transformation groups, including group-equivariant CNNs (Cohen & Welling, 2016; 2017) and equivariant graph networks (Keriven & Peyré, 2019), often improving data efficiency and generalization.

## 3  SOFT EQUIVARIANCE REGULARIZATION FOR INVARIANT SELF-SUPERVISED LEARNING

Previous methods for introducing equivariance into invariant SSL often impose both invariance and equivariance objectives on the *final* representation. However, the final representation is typically spatially collapsed, which is well-suited for invariance but poorly aligned with spatial group actions. SER instead encourages equivariance at an *intermediate spatial representation* that retains a lattice structure, where analytic feature-space actions are naturally defined.

### 3.1  SOFT EQUIVARIANCE AT INTERMEDIATE FEATURES

A straightforward way to introduce equivariance is to impose it directly on the final representation, as explored in prior work. However, these final representations are spatially collapsed and do not admit a natural spatial action. We therefore regularize non-`[CLS]` patch tokens where spatial structure is still explicit (e.g., the gray feature maps in Figure 1).

For ViTs, to preserve a clean spatial token map for equivariance, we decompose the encoder as

$$f = f^{(2)} \circ f^{(1)},$$

where $f^{(1)}$ is a structure-preserving feature extractor that outputs a spatial token map, and $f^{(2)}$ is an invariance-oriented head that produces the final embedding. We introduce the `[CLS]` token only at the input of $f^{(2)}$, so it does not affect the spatial structure of the intermediate representation produced by $f^{(1)}$ (Figure 2).

**Geometric group for equivariance.** Let $\mathcal{G}$ denote the set of *invertible geometric* transformations used by SER (anisotropic scaling without cropping, horizontal flips, and $90°$ rotations). SER defines the input-space and feature-space actions $\rho_g$ only for $g \in \mathcal{G}$.

We adopt the standard two-view SSL protocol (Gupta et al., 2023). Given an image $x$ and two sampled geometric elements $g_1, g_2 \sim \mathcal{G}$, we form two views

$$x_1 = \rho_{g_1}(x), \qquad x_2 = \rho_{g_2}(x),$$

where $\rho_g$ denotes applying the corresponding geometric transform in image space. (We omit photometric augmentations in notation: in practice, both views are also independently subjected to the standard photometric jitter from the base SSL pipeline, but we do not define a feature-space action for these non-group transforms. Consequently, $\mathcal{L}_{\text{equiv}}$ enforces geometric equivariance while being evaluated under potentially different photometric perturbations.) Let $g = g_2 g_1^{-1}$ denote the relative transform such that $x_2 = \rho_g(x_1)$. SER enforces equivariance on the intermediate spatial token map produced by the prefix encoder $f^{(1)}(\cdot)$ by minimizing the equivariance error

$$\mathcal{L}_{\text{equiv}} = \mathbb{E}_{x, g_1, g_2 \sim \mathcal{G}} \Big[ d\big( \rho_g(f^{(1)}(x_1)), \, f^{(1)}(x_2) \big) \Big], \qquad g = g_2 g_1^{-1}, \tag{2}$$

where $d(\cdot, \cdot)$ measures discrepancy between spatial token maps; in this work we instantiate $d$ with a patch-wise contrastive loss (Sec. 3.3). This form of equivariance objective has appeared before (e.g., Eq. 4 in Yu et al. (2024)); our key difference is that we apply it to an intermediate, spatially structured representation and use an analytically specified feature-space action $\rho_g$ directly, without training an additional transformation-prediction module or latent action network. Minimizing $\mathcal{L}_{\text{equiv}}$ does not enforce exact equivariance, but rather encourages *soft* equivariance at that layer.

**Feature-space action.** Since $f^{(1)}(\cdot)$ is defined on a regular patch grid, we implement $\rho_g$ directly on the token map: discrete rotations and flips correspond to *token permutations* on the grid, while scaling is implemented by deterministic grid resampling with the same resize operator as the input-space transform. This lets us apply the known geometric action in feature space without learning an additional action network.

Because $\mathcal{L}_{\text{equiv}}$ alone does not provide an instance-discrimination signal for the final embedding, we jointly train $f^{(2)}$ with a standard invariance-based SSL loss on the final representation (e.g., the `[CLS]` token), as in MoCo-v3, DINO, and Barlow Twins (Chen et al., 2020; He et al., 2020; Grill et al., 2020; Zbontar et al., 2021). Importantly, SER adds no extra transformation/action module (unlike, e.g., EquiMod or STL (Devillers & Lefort, 2023; Yu et al., 2024)).

## 3.2 Augmentation Policy and Batch Partitioning

Typical augmentation policies used in invariance-based SSL include `RandomResizedCrop`, `RandomHorizontalFlip`, and photometric modifications such as color jittering and grayscale. However, `RandomResizedCrop` with cropping does not form a group: the discarded region cannot be recovered by an inverse transform. More importantly, cropping changes the spatial support of the image; as a result, the relative transform $g = g_2 g_1^{-1}$ and the corresponding action on a spatial token map are not well defined.

Therefore, SER splits each mini-batch into two sub-batches (Figure 2). The first sub-batch $b_1$ follows the baseline augmentation policy $\mathcal{T}$ used by the underlying invariant SSL method. The second sub-batch $b_2$ follows a modified *equivariant-view* policy $\mathcal{T}_{\text{eq}}$ that disables cropping while retaining photometric jitter and sampling geometric transforms from the invertible group $\mathcal{G}$:

$$b_1 : \mathcal{T}, \qquad b_2 : \mathcal{T}_{\text{eq}}.$$

Concretely, we define

$$\mathcal{T}_{\text{eq}} = \mathcal{T} \setminus \{\text{Random Crop}\} \cup \{\text{Rotation } 90°\},$$

implemented by replacing `RandomResizedCrop` with its *invertible* geometric component (anisotropic scaling without cropping) and adding discrete $90°$ rotations. Thus, $\mathcal{T}_{\text{eq}}$ retains the full photometric pipeline from $\mathcal{T}$ (color jitter, grayscale, blur, solarization) but samples its invertible geometric transforms from $\mathcal{G}$ (anisotropic scaling without cropping, horizontal flips, and $90°$ rotations). Equivalently, each draw from $\mathcal{T}_{\text{eq}}$ can be decomposed into a geometric transform $g \in \mathcal{G}$ and a

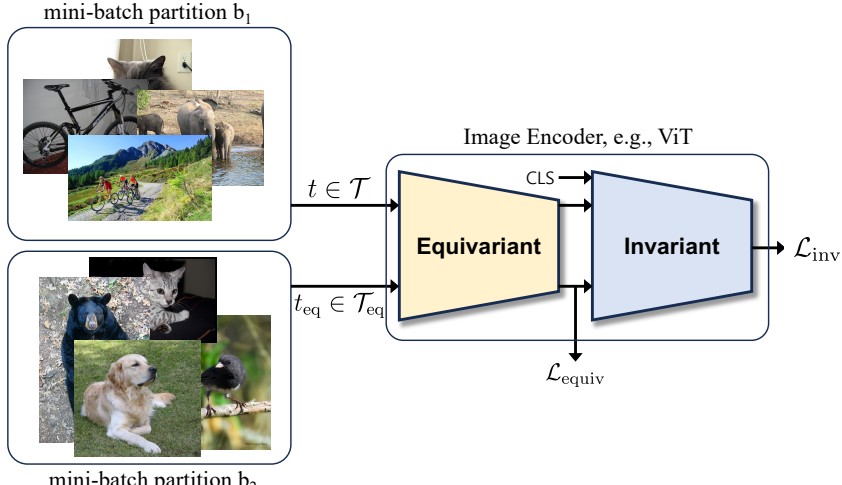

Figure 2: An overview of the training pipeline. The mini-batch is split into $b_1$ and $b_2$: $b_1$ uses the baseline SSL augmentation policy $\mathcal{T}$ (including cropping), while $b_2$ uses an equivariant-view policy $\mathcal{T}_{\text{eq}}$ that disables cropping and adds discrete $90°$ rotations while retaining the baseline photometric jitter. Both $b_1$ and $b_2$ contribute to the baseline invariance loss. SER additionally applies an equivariance regularizer to an intermediate spatial token map using only the *invertible geometric component* of the $b_2$ transforms to define the feature-space action $\rho_g$.

photometric transform $p$. SER uses only $g$ to define its feature-space action $\rho_g$, meaning photometric transformations are *not* elements of the geometric group $\mathcal{G}$ and have no associated feature-space action. Consequently, $\mathcal{L}_{\text{equiv}}$ enforces only geometric equivariance, though optimizing under independently sampled photometric jitter between the two views can additionally encourage photometric robustness at the regularized layer. Both $b_1$ and $b_2$ contribute to the baseline invariant SSL loss; $b_2$ additionally contributes the equivariance regularizer.

**Scaling implementation.** Scaling in $\rho_g$ reuses the same resize operator (interpolation kernel and padding behavior) as the base SSL data pipeline to keep comparisons fair. For ViT-S/16, we restrict scaling factors so that the transformed image remains aligned to the 16-pixel patch grid, ensuring the resulting token map lies on a regular lattice. Rotations by $90°$ and horizontal flips are exact token permutations and require no interpolation.

### 3.3 TRAINING OBJECTIVE FOR SOFT EQUIVARIANCE REGULARIZATION

We instantiate the equivariance regularizer in Equation (2) using a patch-wise NT-Xent (noise-contrastive) loss applied on the sub-batch $b_2$ (Chen et al., 2020). In practice, we reuse the same paired views $(x_1, x_2)$ sampled from $\mathcal{T}_{\text{eq}}$ to compute both the baseline invariant loss $\mathcal{L}_{\text{inv2}}$ and the equivariance regularizer $\mathcal{L}_{\text{equiv}}$; SER therefore does not require additional crops beyond the standard two-view protocol per image.

Let $H_f, W_f, D_f$ denote the height, width, and channel dimension of the intermediate feature maps. For each $x \in b_2$, we sample two augmented views from $\mathcal{T}_{\text{eq}}$ and denote their associated geometric components by $g_1, g_2 \sim \mathcal{G}$; for brevity we write $x_1 = \rho_{g_1}(x)$ and $x_2 = \rho_{g_2}(x)$ and omit the independent photometric jitter applied to each view. We then compute

$$\boldsymbol{z} = \rho_g\big(f^{(1)}(x_1)\big), \qquad \boldsymbol{z}' = f^{(1)}(x_2), \qquad g = g_2 g_1^{-1},$$

with $\boldsymbol{z}, \boldsymbol{z}' \in \mathbb{R}^{H_f \times W_f \times D_f}$. We index images in $b_2$ by $i$ and spatial locations by $j \in \{0, \ldots, H_f W_f - 1\}$, and denote by $z_{ij}$ and $z'_{ij}$ the corresponding feature vectors from $\boldsymbol{z}$ and $\boldsymbol{z}'$, respectively. Each vector is projected by a 2-layer MLP with GELU into a 512-dimensional space (Caron et al., 2021). The equivariance contrastive loss for anchor $(i, j)$ is

$$\mathcal{L}_{\text{equiv}}^{i,j} = -\log \frac{\exp\big(s(z_{ij}, z'_{ij})\big)}{\exp\big(s(z_{ij}, z'_{ij})\big) + \sum_{m \neq i} \sum_n \exp\big(s(z_{ij}, z_{mn})\big) + \sum_{m \neq i} \sum_n \exp\big(s(z_{ij}, z'_{mn})\big)},$$

where $s(x, y)$ denotes temperature-scaled cosine similarity, $s(x, y) = \frac{1}{\tau} x^\top y / (\|x\| \|y\|)$. We set $\tau = 0.3$ for MoCo-v3 and Barlow Twins, and $\tau = 0.5$ for DINO. Following O Pinheiro et al. (2020), negatives are sampled only from *other* images in the batch.

The overall equivariance loss averages this quantity over all images and spatial locations in $b_2$ and contributes to the full training objective, where $\lambda$ controls the strength of equivariance regularization.

$$\mathcal{L} = \mathcal{L}_{\text{inv1}} + \mathcal{L}_{\text{inv2}} + \lambda \mathcal{L}_{\text{equiv}}, \qquad \mathcal{L}_{\text{equiv}} = \frac{1}{|b_2| H_f W_f} \sum_i \sum_j \mathcal{L}_{\text{equiv}}^{i,j}, \tag{3}$$

Both $\mathcal{L}_{\text{inv1}}$ and $\mathcal{L}_{\text{inv2}}$ use exactly the baseline invariance-based SSL loss (e.g., MoCo-v3, DINO, Barlow Twins) applied to sub-batches $b_1$ and $b_2$, respectively. Thus, SER is agnostic to the choice of base SSL algorithm and can be seamlessly integrated with different invariance-based methods.

## 4 EXPERIMENTS

We empirically evaluate *Soft Equivariance Regularization* (SER) against representative equivariant SSL baselines and strong invariant SSL backbones. We first describe the experimental setup (Section 4.1) and then address the following questions:

- Does SER improve ViT representations compared to purely invariant SSL and prior equivariant SSL add-ons?
- Does SER scale to ImageNet-1k pretraining, and how does it impact robustness and spatially sensitive transfer?
- Which design choices are responsible for the gains (layer placement, [CLS] insertion, and the equivariance loss itself)?

### 4.1 EXPERIMENTAL SETUP

**Baselines.** We compare SER with representative equivariant SSL approaches, including implicit methods (E-SSL (Dangovski et al., 2021), AugSelf (Lee et al., 2021)) and explicit methods (EquiMod (Devillers & Lefort, 2023), STL (Yu et al., 2024)). Because several baselines use different numbers of views (e.g., EquiMod: 3 global crops; E-SSL: 2 global + 4 local crops), direct comparisons can be confounded by view count. Since increasing the number of views typically improves SSL performance but increases compute and memory (Caron et al., 2020; 2021), we emphasize *matched-view* comparisons. In addition to the standard 2-view setting, we also report SER under a matched 2+4 scheme for fair comparison with methods that use local crops.

**Datasets and evaluation benchmarks.** We pretrain and evaluate on ImageNet-1k (Deng et al., 2009), following standard SSL protocols (Chen et al., 2020; Caron et al., 2021). We additionally evaluate generalization under natural distribution shifts (ImageNet-V2 (Recht et al., 2019), ImageNet-R (Hendrycks et al., 2021), ImageNet-Sketch (Wang et al., 2019)) and robustness under corruptions/perturbations (ImageNet-C and ImageNet-P (Hendrycks & Dietterich, 2019)). ImageNet-P emphasizes geometric perturbations (e.g., translation, rotation, and scaling), whereas ImageNet-C primarily focuses on appearance-based corruptions (e.g., blur, noise, brightness/fog). We also evaluate out-of-domain transfer on 3DIEBench (Garrido et al., 2023) via linear probing on its 2D renderings (we do not exploit or supervise 3D transformation parameters), and evaluate spatial transfer with frozen-backbone COCO object detection.

**Implementation details.** Unless otherwise noted, we pretrain ViT-S/16 on ImageNet-1k. SER is implemented as an add-on to invariant SSL backbones (MoCo-v3 (Chen et al., 2021), DINO (Caron et al., 2021), Barlow Twins (Zbontar et al., 2021)); we preserve their architectures and default hyperparameters. Our modifications are limited to: (i) partitioning each mini-batch into $b_1$ (baseline augmentations) and $b_2$ (group-compatible augmentations for equivariance); (ii) inserting the [CLS] token *after* the equivariance-regularized layer to preserve a spatial token map; and (iii) adding the equivariance regularizer and its projection MLP.

Table 1: **Top-1 and top-5 accuracy (%) under linear evaluation.** Note that all equivariant representation learning methods use MoCo (He et al., 2020) as their baseline, which outperformed DINO and BarlowTwins in our setting (see Table 2). Concatenated [CLS] tokens from the last 4 layers were used as an input to the linear classifier, following the feature-based evaluations in (Devlin et al., 2019; Caron et al., 2021). 'View' refers to the number of crops sampled per image (see Section A for more detail). ImageNet-1k scores are averaged over 3 runs.

| View | Algorithm | Param (M) | ImageNet-1k | | ImageNet-Sketch | | ImageNet-V2 | | ImageNet-R | | 3DIEBench | |
|---|---|---|---|---|---|---|---|---|---|---|---|---|
| | | | Top-1 | Top-5 | Top-1 | Top-5 | Top-1 | Top-5 | Top-1 | Top-5 | Top-1 | Top-5 |
| 2 view | MoCo-v3 | 42.9 | 68.44 ±0.07 | 88.02 ±0.04 | 17.65 | 31.87 | 56.54 | 78.68 | 18.59 | 30.08 | 68.43 | 91.96 |
| | + AugSelf | 43.7 | 67.55 ±0.05 | 87.62 ±0.05 | 13.30 | 25.35 | 53.74 | 76.68 | 17.62 | 28.66 | 64.97 | 90.73 |
| | + STL | 62.2 | 65.49 ±0.12 | 85.91 ±0.08 | 15.40 | 28.96 | 55.43 | 78.02 | 17.22 | 28.49 | - | - |
| | + SER | 43.4 | **69.28 ±0.01** | **88.79 ±0.02** | **17.68** | **32.54** | **56.95** | **79.29** | **18.95** | **30.72** | **70.17** | **92.78** |
| 3 view | + EquiMod | 43.3 | 68.95 ±0.02 | 88.87 ±0.01 | 14.81 | 28.11 | 56.31 | 79.93 | 16.54 | 27.32 | 67.97 | 91.97 |
| 2+4 view | + E-SSL | 43.3 | 70.6 ±0.04 | 89.85 ±0.02 | 19.23 | 34.77 | 58.33 | **80.93** | 19.86 | 32.36 | - | - |
| | + SER | 43.4 | **71.56 ±0.03** | **90.04 ±0.01** | **19.76** | **34.81** | **59.50** | 80.72 | **20.27** | **32.54** | **70.91** | **93.15** |

Table 2: **Top-1 and top-5 accuracy (%) under linear evaluation with different baseline invariant self-supervised learning (SSL) methods.** All methods use 2-view augmentation policy, and ImageNet-1k scores are averaged over 3 runs.

| Algorithm | ImageNet-1k | | ImageNet-Sketch | | ImageNet-V2 | | ImageNet-R | |
|---|---|---|---|---|---|---|---|---|
| | Top-1 | Top-5 | Top-1 | Top-5 | Top-1 | Top-5 | Top-1 | Top-5 |
| MoCo-v3 | 68.44 ±0.07 | 88.02 ±0.04 | 17.65 | 31.87 | 56.54 | 78.68 | 18.59 | 30.08 |
| + SER | **69.28 ±0.01** | **88.79 ±0.02** | **17.68** | **32.54** | **56.95** | **79.29** | **18.95** | **30.72** |
| DINO | 67.37 ±0.02 | 87.55 ±0.01 | 17.13 | 32.09 | 55.00 | 77.38 | 18.28 | 30.38 |
| + SER | **67.63 ±0.01** | **87.56 ±0.01** | **18.07** | **34.03** | **55.19** | **77.84** | **18.96** | **31.55** |
| Barlow Twins | 63.34 ±0.03 | 84.3 ±0.04 | 10.90 | 21.17 | 47.69 | 70.73 | 12.30 | 20.94 |
| + SER | **64.02 ±0.03** | **84.73 ±0.01** | **12.39** | **24.39** | **50.89** | **74.20** | **13.90** | **23.99** |

For all studies, we pretrain with AdamW using batch size 2048 for 100 epochs, with 10-epoch warmup and cosine learning-rate decay. We use a scaling range of $[0.7, 1.3]$ in the geometric pipeline. For linear evaluation, we concatenate [CLS] tokens from the last 4 layers as input to a linear classifier (Devlin et al., 2019; Caron et al., 2021) and train for 50 epochs with cosine decay (no warmup), following common ViT SSL practice. (Where reported, we use multi-seed evaluation and provide mean $\pm$ std.)

## 4.2 MAIN RESULTS

**Linear evaluation on ImageNet-1k.** We evaluate representation quality via linear probing on ImageNet-1k. As shown in Table 1, SER improves over the invariant baselines and compares favorably to prior equivariant add-ons. Because equivariant baselines use different numbers of views, we emphasize matched-view comparisons. In the strictly matched 2-view regime, several equivariant add-ons reduce ImageNet-1k accuracy relative to MoCo-v3, whereas SER improves MoCo-v3 by **+0.84** Top-1. We additionally report SER in a matched 2+4-view setting for fair comparison with methods that use local crops.

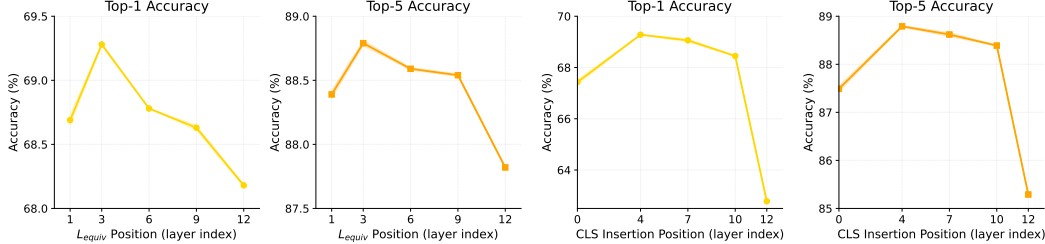

Figure 3: Ablation study on the location to regularize towards equivariance (left) and to insert the `[CLS]` token in the ViT encoder with fixed equivariance regularization layer at the 3rd layer (right). Both Top-1 (left) and Top-5 (right) accuracies peak when the equivariance loss and `[CLS]` is introduced near the middle of the network.

Table 3: **Nonlinear evaluations using ImageNet-1k with different equivariant representation learning methods.** Note that all equivariant representation learning methods use MoCo as their baseline. View refers to the number of crops sampled per image.

| View | Algorithm | MLP | | 20-NN | Fine-tune | |
|---|---|---|---|---|---|---|
| | | Top-1 | Top-5 | Top-1 | Top-1 | Top-5 |
| 2 view | MoCo-v3 | 67.84 | **88.37** | 61.56 | 73.83 | **91.91** |
| | + AugSelf | 63.24 | 85.86 | 60.63 | 73.50 | 91.67 |
| | + STL | 65.74 | 86.32 | 57.34 | 73.90 | 91.49 |
| | + SER | **68.04** | **88.37** | **61.64** | **74.33** | 91.79 |
| 3 view | + EquiMod | 68.33 | 88.50 | 58.28 | 74.08 | 91.75 |
| 2+4 view | + E-SSL | 68.45 | 88.31 | 64.56 | 75.00 | **92.36** |
| | + SER | **70.99** | **89.72** | **65.32** | **75.02** | 92.12 |

**Robustness and spatial transfer.** On MoCo-v3 (ViT-S/16, matched 2-view), SER improves average Top-1 on ImageNet-C/P by **+1.11/+1.22** and frozen-backbone COCO detection by **+1.7** mAP (Appendix Sections A.3 and A.5, Tables 11 to 13).

**Generalizability across invariant SSL methods.** Although MoCo-v3 is our main backbone, we also evaluate SER on DINO (Caron et al., 2021) and Barlow Twins (Zbontar et al., 2021). As shown in Table 2, SER consistently improves all baselines without architecture changes.

**Nonlinear evaluation and fine-tuning.** Following (Garrido et al., 2023), we evaluate learned representations under nonlinear probing (3-layer MLP), $k$-NN evaluation, and full fine-tuning. Results are reported in Table 3.

## 4.3 ABLATION AND ANALYSIS

A key claim of SER is that equivariance should be enforced on an *intermediate* spatial representation, whereas many prior approaches impose both invariance and equivariance on the final representation (Lee et al., 2021; Dangovski et al., 2021; Devillers & Lefort, 2023; Yu et al., 2024). We also insert the `[CLS]` token *after* the equivariance-regularized layer to preserve spatial structure (Figure 2). Figure 3 shows an intermediate "sweet spot" for both the equivariance-loss layer and the `[CLS]` insertion location.

We further examine the relationship between equivariance strength and discriminative quality when moving the equivariance regularizer closer to the output. As shown in Table 4, deeper equivariance regularization increases equivariance scores measured at the final layer, but steadily reduces ImageNet-1k linear-eval accuracy.

**Control: $\lambda = 0$ (isolating the equivariance loss).** To isolate the effect of $\mathcal{L}_{\text{equiv}}$ from batch partitioning and augmentation changes, we train a control model with identical settings but set $\lambda =$

Table 4: **Layer-wise ablation for the effect of equivariance regularization.** Moving $\mathcal{L}_{\text{equiv}}$ toward deeper layers increases the equivariance score of the final representation but reduces ImageNet-1k linear-eval accuracy, indicating a trade-off when equivariance is enforced closer to the final representation. Higher ↑ indicates greater equivariance.

| Algorithm | Equivariance Loss Layer | ImageNet | Rotation ↑ | Hflip ↑ | Scale ↑ |
|---|---|---|---|---|---|
| MoCo-v3 | - | 68.44 | 0.804 | 0.870 | 0.945 |
| MoCo + SER | Layer 3 | **69.28** | 0.840 | 0.881 | 0.945 |
| MoCo + SER | Layer 9 | 68.72 | 0.888 | 0.886 | **0.957** |
| MoCo + SER | Layer 12 | 68.18 | **0.924** | **0.892** | **0.957** |

Table 5: Control experiment isolating the effect of the equivariance regularizer (3 seeds).

| Method | ImageNet-1k Top-1 (%) |
|---|---|
| MoCo-v3 (baseline) | $68.44 \pm 0.07$ |
| +SER, $\lambda = 0$ (control) | $68.82 \pm 0.01$ |
| +SER, $\lambda > 0$ (full) | $\mathbf{69.28 \pm 0.01}$ |

Table 6: Effect of applying layer-decoupling recipe to prior invariant+equivariant baselines using MoCo-v3 by moving equivariance objective from the final to an intermediate layer (3-seeds average).

| Method | Equiv. loss layer | ImageNet-1k Top-1 (%) |
|---|---|---|
| EquiMod (Devillers & Lefort, 2023) | 12 (final) | $68.95 \pm 0.02$ |
| EquiMod (Devillers & Lefort, 2023) | 3 (intermediate) | $\mathbf{69.51 \pm 0.02}$ |
| AugSelf (Lee et al., 2021) | 12 (final) | $67.55 \pm 0.05$ |
| AugSelf (Lee et al., 2021) | 3 (intermediate) | $\mathbf{68.23 \pm 0.06}$ |

0. As shown in Table 5, batch/augmentation changes account for a modest gain, while enabling $\mathcal{L}_{\text{equiv}}$ yields an additional, statistically stable improvement.

**Layer decoupling upgrades prior invariant+equivariant methods.** To test whether "equivariance-at-an-intermediate-layer" is a general design principle, we re-implement EquiMod and AugSelf with their equivariance losses moved from the final layer to the same intermediate layer used by SER, keeping the rest of each method unchanged. Table 6 shows that both methods improve, suggesting that layer decoupling can strengthen existing invariant+equivariant approaches.

**Additional ablations.** We report sensitivity to the partition ratio $r$ and loss weight $\lambda$, group-element removal, and applying $\mathcal{L}_{\text{equiv}}$ to all layers in the appendix (Sec. A).

## 5 CONCLUSION

We introduced *Soft Equivariance Regularization* (SER), a simple and scalable way to incorporate equivariance into invariant self-supervised learning. SER decouples invariance and equivariance across layers: the final representation is trained with an unchanged baseline SSL objective, while an intermediate spatial token map is softly regularized toward equivariance via analytically specified group actions.

Empirically, we show that coupling invariance and equivariance on the same final (often spatially collapsed) representation can be suboptimal, and that intermediate-layer regularization yields a better trade-off. SER avoids auxiliary transformation/action modules and does not require *learning or predicting* per-sample transformation labels/codes. Across multiple invariant SSL baselines for ViTs pretrained on ImageNet-1k, SER consistently improves downstream accuracy and robustness, with minimal additional compute. Finally, our results suggest that *layer decoupling* is a useful design principle for combining invariance and equivariance, and can also strengthen existing invariant+equivariant methods when applied to their equivariance objectives.

## ACKNOWLEDGEMENT

This work was partly supported by Institute of Information & communications Technology Planning & Evaluation (IITP) grant funded by the Korea government (MSIT) (No.RS-2019-II190075, Artificial Intelligence Graduate School Program (KAIST)) and Institute of Information & communications Technology Planning & Evaluation (IITP) grant funded by the Korea government (MSIT) (No.RS-2024-00509279, Global AI Frontier Lab).

## ETHICS STATEMENT

All authors acknowledge and commit to complying with the ICLR Code of Ethics. Our work proposes Soft Equivariance Regularization (SER), a methodological contribution to self-supervised visual representation learning, and does not involve human-subject experimentation, user studies, or the collection of new personal data. All experiments use only publicly available datasets (ImageNet-1k, MS COCO, 3DIEBench, ImageNet-R, ImageNet-Sketch, ImageNet-V2, ImageNet-C, and ImageNet-P) under their respective licenses/terms. We recognize that widely used vision benchmarks can contain societal and demographic biases and may include images of individuals; consequently, models trained with SER may inherit or amplify such biases when deployed downstream. We therefore encourage practitioners to perform domain-appropriate bias/fairness, privacy, and misuse-risk evaluations (e.g., surveillance-related applications) prior to deployment, and to follow applicable legal and licensing constraints. To support research integrity and transparency, we report matched-view comparisons, document limitations, and provide code for independent verification.

## REPRODUCIBILITY STATEMENT

To support reproducibility, all experimental results reported in this paper are produced from the public implementation at `https://github.com/aitrics-chris/SER`. The SER method is fully specified in the main paper (including the training pipeline, augmentation/batch partitioning, analytic feature-space group actions, and the objective; see Section 3 and Algorithm 1), while the experimental protocols and key hyperparameters are described in Section 4.1, with additional ablations and benchmark-specific details (e.g., matched-view settings, robustness/transfer evaluations, and sensitivity to $\lambda$ and batch partition ratio) provided in the Section A. Multi-seed performance numbers are computed by averaging results over 3 random seeds (and we report variability where applicable). All evaluations use standard public benchmarks with established splits and metrics, enabling results to be reproduced directly by running the released training/evaluation scripts and configurations.

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

---

**Algorithm 1** Soft Equivariance Regularization (SER) for invariant self-supervised learning

---

1: **Input:** batch $B$, partition ratio $r$, baseline augmentation policy $\mathcal{T}$, equivariant-view policy $\mathcal{T}_{\mathrm{eq}}$ (crop disabled; photometric jitter retained), geometric group $\mathcal{G}$ (scale, HFlip, Rot90°), encoder $f = f^{(2)} \circ f^{(1)}$, base SSL loss $\mathcal{L}_{\mathrm{SSL}}(\cdot)$, weight $\lambda$
2: Partition $B$ into $b_1$ and $b_2$ where $|b_2| = r|B|$ and $|b_1| = (1-r)|B|$
   *// Sample paired views for the baseline SSL loss*
3: **for** each $x \in b_1$ **do**
4:     Sample two views $(x_1, x_2) \sim \mathcal{T}$                          ▷ baseline 2-view pipeline
5: **end for**
6: **for** each $x \in b_2$ **do**
7:     Sample two views $(x_1, x_2) \sim \mathcal{T}_{\mathrm{eq}}$ and record geometric parts $g_1, g_2 \in \mathcal{G}$     ▷ photometric jitter in $\mathcal{T}_{\mathrm{eq}}$ is sampled independently across the two views
8: **end for**
   *// Invariance losses (on the final embedding, unchanged)*
9: $\mathcal{L}_{\mathrm{inv1}} \leftarrow \mathcal{L}_{\mathrm{SSL}}(b_1; \mathcal{T})$
10: $\mathcal{L}_{\mathrm{inv2}} \leftarrow \mathcal{L}_{\mathrm{SSL}}(b_2; \mathcal{T}_{\mathrm{eq}})$
   *// Equivariance regularizer on intermediate spatial features (reusing the same $b_2$ views)*
11: Compute intermediate token maps $\{h_1^{(i)}\} \leftarrow f^{(1)}(\{x_1^{(i)}\})$ and $\{h_2^{(i)}\} \leftarrow f^{(1)}(\{x_2^{(i)}\})$ for all $x^{(i)} \in b_2$
12: **for** each $i$ with $x^{(i)} \in b_2$ **do**
13:     Relative geometric transform: $g^{(i)} \leftarrow g_2^{(i)}(g_1^{(i)})^{-1}$
14:     Align features: $\hat{h}_1^{(i)} \leftarrow \rho_{g^{(i)}}(h_1^{(i)})$
15: **end for**
16: $\mathcal{L}_{\mathrm{equiv}} \leftarrow \texttt{PatchNTXent}\big(\{\hat{h}_1^{(i)}\}_{i \in b_2}, \{h_2^{(i)}\}_{i \in b_2}\big)$     ▷ negatives from other images/tokens
   *// Total objective*
17: $\mathcal{L} \leftarrow \mathcal{L}_{\mathrm{inv1}} + \mathcal{L}_{\mathrm{inv2}} + \lambda \mathcal{L}_{\mathrm{equiv}}$
18: Update encoder parameters by minimizing $\mathcal{L}$
19: **Output:** pre-trained encoder $f$

---

# A   FURTHER DISCUSSIONS AND EXPERIMENTS

## A.1   NUMBER OF VIEWS AND MATCHED-VIEW COMPARISONS

It is well established that increasing the number of global/local views (augmentations) can improve SSL representation quality, at the cost of additional compute and memory (Caron et al., 2020; 2021). Therefore, comparing methods that use different numbers of views can be confounded by view count. In particular, E-SSL uses a 2+4-view strategy (2 global and 4 local views), while EquiMod uses 3 global views.

To ensure fair comparisons, we also implement SER under a matched 2+4-view setting. Following the local-to-global design in DINO (Caron et al., 2021), we do not pass all four local views through the MoCo momentum encoder, which avoids computing losses among local views. For the equivariance regularizer, we form one global pair and two local pairs, and compute the equivariance loss within each pair. Results of this 2+4-view variant are reported separately under the "2+4-view" rows in the tables.

## A.2   ADDITIONAL ABLATIONS AND ANALYSIS

**Equivariance-loss layer and `[CLS]` insertion.**   In Section 4.3, we evaluate the common practice of imposing equivariance and invariance objectives at the encoder's final representation. Our results show that applying the equivariance regularizer too early or too late is suboptimal, while the best downstream performance is achieved by regularizing an intermediate layer (in our main ViT-S/16 setting, the third transformer block), where spatial structure is preserved. We also find that the `[CLS]` insertion point matters: inserting `[CLS]` too early can hinder equivariance regularization on spatial tokens, while inserting it too late can degrade invariance learning (Figure 3).

Table 7: **Transformation prediction.** Evaluation of transformation label prediction from the learned representation of different layers (see Section A for more detail)

| Tasks | Methods | Layer 1 | Layer 3 | Layer 6 | Layer 9 | Layer 12 |
|---|---|---|---|---|---|---|
| Rotation Prediction (%) | MoCo + AugSelf | **81.55** | **96.71** | **99.71** | 99.13 | 68.74 |
| | MoCo + STL | 79.56 | 92.8 | 99.46 | **99.76** | **98.08** |
| | MoCo + SER | 79.97 | 93.34 | 99.52 | 99.73 | 96.59 |
| HFlip Prediction (%) | MoCo + AugSelf | **63.24** | **85.66** | **96.48** | 91.51 | 62.68 |
| | MoCo + STL | 62.91 | 82.09 | 88.83 | **97.22** | **83.27** |
| | MoCo + SER | 63.02 | 78.68 | 92.87 | 96.59 | 75.95 |

Table 8: Sensitivity to the batch partition ratio $r$ and equivariance loss weight $\lambda$ (ImageNet-1k linear eval, 3 seeds).

| $r$ | 0 (MoCo-v3) | 0.005 | 0.01 | 0.02 |
|---|---|---|---|---|
| Top-1 (%) | $68.44 \pm 0.07$ | $69.08 \pm 0.03$ | $\mathbf{69.28 \pm 0.01}$ | $68.30 \pm 0.05$ |

| $\lambda$ | 0 (control) | 0.5 | 1.0 | 5.0 | 10.0 |
|---|---|---|---|---|---|
| Top-1 (%) | $68.82 \pm 0.01$ | $\mathbf{69.28 \pm 0.01}$ | $68.92 \pm 0.02$ | $68.73 \pm 0.01$ | $68.67 \pm 0.03$ |

**Equivariance score.** In Table 4, we quantify how equivariance measured at the *final* representation changes when shifting the equivariance-loss layer. Following (Zhang, 2019), we sample geometric transforms from rotation 90°, horizontal flip, and scaling, and measure an equivariance score using cosine similarity:

$$\text{Equivariance} = \mathbb{E}_{x,\,(g_1,g_2)\sim\mathcal{G}}\Big[\cos\big(\rho_g(f(x)),\ f(\rho_g(x))\big)\Big], \qquad g = g_2 g_1^{-1},$$

where $f(\cdot)$ denotes the final representation (thus we evaluate equivariance at the last layer). Higher values indicate stronger equivariance.

**Transformation prediction probes.** In Table 7, we evaluate whether frozen features support predicting rotation/flip labels, following (Garrido et al., 2023). This is a classification accuracy (not an $R^2$ regression score). Although SER regularizes equivariance at an intermediate layer, deeper layers (including the final layer) retain strong transformation information.

**Sensitivity to $r$ and $\lambda$.** We evaluate sensitivity to the batch partition ratio $r$ and the equivariance loss weight $\lambda$ (ImageNet-1k linear eval, 3 seeds). SER remains stable across a reasonable range of values and does not rely on finely tuned hyperparameters.

**Ablating geometric group elements.** To assess sensitivity to the choice of invertible geometric transformations, we ablate elements of the geometric set used by SER. Performance remains competitive and above the MoCo-v3 baseline as long as a reasonable geometric group is available.

**Applying $\mathcal{L}_{\text{equiv}}$ on all layers.** We compare applying the equivariance regularizer to a single intermediate layer versus all transformer blocks. Regularizing all layers provides gains, but is slightly worse than the best single-layer choice and increases compute.

## A.3 FROZEN-BACKBONE OBJECT DETECTION ON COCO

Equivariance is expected to be particularly helpful for tasks that require spatial sensitivity beyond image-level classification. To further examine transfer, we evaluate frozen-encoder object detection on COCO Lin et al. (2014). As shown in Table 11, SER achieves the strongest detection performance among the compared pretrained encoders under the same detection setup. Following Oquab et al. (2023), we freeze encoder weights and train the detection head for 45k iterations with batch size 32 on COCO2017 train, reporting performance on COCO2017 val. All methods share the same detection pipeline; only the pretrained encoder differs.

Table 9: Ablating geometric group elements used in SER (3 seeds).

| Variant | ImageNet-1k Top-1 (%) |
|---|---|
| SER (full) | $\mathbf{69.28 \pm 0.01}$ |
| SER – Rot90° | $69.17 \pm 0.01$ |
| SER – Rot90° – HFlip | $69.12 \pm 0.12$ |
| SER – Rot90° – Scale | $68.67 \pm 0.01$ |

Table 10: Applying $\mathcal{L}_{\text{equiv}}$ to all transformer blocks vs. a single intermediate layer (3 seeds).

| Variant | Top-1 (%) | Top-5 (%) |
|---|---|---|
| MoCo-v3 | $68.44 \pm 0.07$ | $88.02 \pm 0.04$ |
| +SER (layer 3) | $\mathbf{69.28 \pm 0.01}$ | $\mathbf{88.79 \pm 0.02}$ |
| +SER (all layers) | $69.21 \pm 0.05$ | $88.78 \pm 0.12$ |

Table 11: COCO object-detection results with a frozen backbone (higher is better).

| Metric | MoCo | MoCo + SER | MoCo + STL | MoCo + AugSelf |
|---|---|---|---|---|
| mAP | 0.225 | **0.242** | 0.221 | 0.197 |
| mAP@50 | 0.404 | **0.428** | 0.400 | 0.359 |
| mAP@75 | 0.222 | **0.244** | 0.218 | 0.192 |

Table 12: Top-1 accuracy comparison on ImageNet-C, including 15 types of common corruptions, for our method and other equivariant representation learning methods built upon the invariant representation learning baseline MoCo (He et al., 2020).

| Algorithm | Noise | | | Blur | | | | Weather | | | | Digital | | | | Avg. |
|---|---|---|---|---|---|---|---|---|---|---|---|---|---|---|---|---|
| | Gauss. | Shot | Impul. | Defo. | Glass | Motion | Zoom | Snow | Frost | Fog | Bright. | Cont. | Elas. | Pixel | JPEG | |
| MoCo-v3 | 39.18 | 37.81 | 36.09 | 33.51 | 13.85 | 31.49 | 25.86 | 30.61 | 30.03 | 35.02 | 62.81 | 52.00 | 54.65 | 55.78 | 53.37 | 39.47 |
| + AugSelf | 34.91 | 31.81 | 31.44 | 35.06 | **17.12** | 34.28 | **27.67** | **31.99** | 28.50 | 35.01 | 61.99 | 50.64 | 55.48 | 54.88 | 53.17 | 38.93 |
| + STL | 17.78 | 16.26 | 14.65 | 29.51 | 15.13 | 27.33 | 25.77 | 29.50 | 27.60 | 34.40 | 61.81 | 48.84 | 54.31 | 46.40 | 50.63 | 33.33 |
| + SER | 39.42 | 38.30 | 36.85 | 36.23 | 15.91 | **34.90** | 27.35 | 30.71 | 30.12 | 36.04 | 63.60 | 52.81 | 55.85 | 56.63 | 54.03 | 40.58 |
| + EquiMod† | 34.33 | 32.59 | 31.95 | 31.81 | 15.37 | 31.76 | 27.38 | 29.08 | 25.73 | 31.38 | 61.94 | 47.41 | 55.07 | 53.58 | 52.40 | 37.45 |
| + E-SSL‡ | **43.80** | **42.59** | **40.80** | **38.44** | 16.40 | **36.73** | 28.20 | 34.22 | 32.24 | 37.93 | 65.50 | **55.89** | 56.12 | 55.80 | 55.21 | 42.66 |
| + SER‡ | 39.88 | 39.27 | 37.18 | 36.21 | **19.58** | 34.16 | **31.90** | **36.07** | **35.87** | **40.74** | **66.76** | 54.90 | **57.87** | 56.42 | 56.79 | **42.91** |

## A.4 WHY SER DOES NOT COLLAPSE TO A TRIVIAL SOLUTION

We briefly explain why SER does not collapse to a trivial intermediate representation. Minimizing $\mathcal{L}_{\text{equiv}}$ corresponds to minimizing $d(\rho_g(f^{(1)}(x)), f^{(1)}(\rho_g(x)))$. First, a trivially invariant intermediate representation does not generally yield zero loss: if $f^{(1)}(\rho_g(x)) = f^{(1)}(x)$, then the loss becomes $d(\rho_g(f^{(1)}(x)), f^{(1)}(x))$, which is nonzero unless $f^{(1)}(x)$ is itself invariant under $\rho_g$ (e.g., a spatially constant map). Second, our contrastive $\mathcal{L}_{\text{equiv}}$ discourages collapse by promoting separation from negatives sampled from other images and spatial locations, which encourages diversity rather than spatial constancy (cf. (Wang & Isola, 2020)). Third, joint optimization with the baseline invariance objective $\mathcal{L}_{\text{inv}}$ (e.g., MoCo) further promotes non-trivial, instance-discriminative features.

## A.5 ADDITIONAL ROBUSTNESS RESULTS ON IMAGENET-C AND IMAGENET-P

This section reports the ImageNet-C and ImageNet-P results referenced in the abstract and the main text (Sec. 4). For MoCo-v3 (ViT-S/16, matched 2-view), SER improves average Top-1 on ImageNet-C from 39.47 to 40.58 (**+1.11**, Table 12) and on ImageNet-P from 63.91 to 65.13 (**+1.22**, Table 13). We additionally report ImageNet-P results when applying SER to other invariant SSL backbones in Table 14.

Table 13: Top-1 accuracy comparison on ImageNet-P, including 14 perturbation types, for our method and other equivariant representation learning methods built upon the invariant representation learning baseline MoCo (He et al., 2020).

| Algorithm | Noise | | | Blur | | | Weather | | | Digital | | | | | Avg. |
|---|---|---|---|---|---|---|---|---|---|---|---|---|---|---|---|
| | Gau. N. | Shot | Speck. | Motion | Zoom | Gau. B. | Snow | Spatter | Bright. | Trans. | Rot. | Tilt | Scale | Shear | |
| MoCo-v3 | 67.85 | 67.85 | 67.97 | 57.31 | 68.30 | 68.28 | 56.57 | 66.57 | 63.43 | 68.05 | 64.55 | 67.58 | 45.18 | 65.32 | 63.91 |
| + AugSelf | 67.44 | 67.47 | 67.47 | 57.51 | 67.76 | 67.81 | 56.77 | 66.15 | 60.77 | 67.39 | 64.41 | 67.10 | 47.15 | 64.96 | 63.58 |
| + STL | 66.19 | 66.14 | 66.15 | 54.97 | 66.29 | 66.32 | 54.38 | 64.71 | 61.16 | 66.00 | 62.07 | 65.86 | 42.53 | 63.17 | 61.85 |
| + SER | **68.97** | **69.01** | **69.00** | 59.09 | **69.46** | **69.35** | 58.02 | 67.86 | 64.58 | 69.04 | 65.76 | 68.60 | 46.78 | **66.34** | **65.13** |
| + EquiMod[†] | 68.60 | 68.70 | 68.78 | 57.08 | 69.17 | 69.13 | 56.97 | 67.51 | 60.88 | 68.75 | 65.18 | 68.27 | 47.04 | 66.17 | 64.44 |
| + E-SSL[‡] | 70.26 | 70.19 | 70.22 | **61.49** | 70.65 | 70.54 | 60.60 | 68.87 | 65.82 | 70.27 | 66.92 | 69.82 | 48.86 | 67.59 | 66.58 |
| + SER[‡] | **71.68** | **71.65** | **71.67** | 60.36 | **71.87** | **71.87** | 61.92 | 70.47 | 67.18 | 71.62 | 68.62 | 71.34 | 52.74 | 68.99 | **68.00** |

Table 14: Experiments with various SSL algorithms. Top-1 accuracy (%) on **ImageNet-P**. All models are trained with the setting addressed in Section 4.1. See Table 13 for the results from MoCo.

| Algorithm | Noise | | | Blur | | | Weather | | Digital / Geometric | | | | | | Avg. |
|---|---|---|---|---|---|---|---|---|---|---|---|---|---|---|---|
| | G.Nse | Shot | Spkl | Mot. | Zoom | G.Blr | Snow | Spat | Brt. | Tran | Rot | Tilt | Scal | Shear | |
| DINO | 66.69 | 66.67 | 66.70 | 52.74 | 67.02 | 66.94 | 55.83 | 65.60 | 61.27 | 66.70 | 62.73 | 66.49 | **42.96** | 63.51 | 62.27 |
| + SER | **67.39** | **67.31** | **67.41** | 54.41 | **67.69** | **67.66** | 56.19 | 66.06 | 62.13 | 67.22 | 63.33 | 67.02 | 42.76 | 64.31 | **62.92** |
| Barlow Twins | 60.09 | 60.08 | 60.12 | 42.25 | 60.53 | 60.53 | 46.37 | 58.84 | 52.92 | 60.22 | 55.47 | 59.37 | 33.15 | 56.79 | 54.77 |
| + SER | **63.85** | **63.93** | **63.91** | 49.18 | **64.29** | **64.19** | 50.09 | 62.34 | 57.93 | 63.89 | 59.63 | 63.38 | 37.29 | 60.56 | **58.89** |

Table 15: Top-5 accuracy comparison on ImageNet-C, including 15 types of common corruptions, for our method and other equivariant representation learning methods built upon the invariant representation learning baseline MoCo (He et al., 2020).

| Algorithm | Noise | | | Blur | | | | Weather | | | | Digital | | | | Avg. |
|---|---|---|---|---|---|---|---|---|---|---|---|---|---|---|---|---|
| | Gauss. | Shot | Impul. | Defo. | Glass | Motion | Zoom | Snow | Frost | Fog | Bright. | Cont. | Elas. | Pixel | JPEG | |
| MoCo-v3 | 63.30 | 61.68 | 59.76 | 56.22 | 28.46 | 53.52 | 45.98 | 52.52 | 50.54 | 58.64 | 84.47 | 76.74 | 77.38 | 79.44 | 77.76 | 61.76 |
| + AugSelf | 58.96 | 55.21 | 54.61 | 59.06 | 34.52 | 57.55 | **48.71** | **55.05** | 49.48 | 59.30 | 84.39 | 76.26 | 78.25 | 79.15 | 78.02 | 61.90 |
| + STL | 37.10 | 34.36 | 32.04 | 52.71 | 31.08 | 48.45 | 46.35 | 50.96 | 47.60 | 59.05 | 84.07 | 74.74 | 76.96 | 71.37 | 75.66 | 54.83 |
| + SER | **64.16** | **62.70** | **61.21** | 59.92 | 32.28 | 58.17 | 48.22 | 53.23 | **51.26** | 60.49 | 85.68 | **77.94** | 78.55 | 80.52 | 78.58 | **63.53** |
| + EquiMod[†] | 58.63 | 56.28 | 55.34 | 55.28 | 31.48 | 54.45 | 48.20 | 51.11 | 45.88 | 55.49 | 84.63 | 73.66 | 78.51 | 78.46 | 77.62 | 60.33 |
| + E-SSL[‡] | **68.70** | **67.42** | **65.64** | **63.00** | 33.09 | **60.35** | 49.83 | 57.64 | 53.72 | 62.83 | 86.80 | **80.33** | 79.05 | **80.32** | 80.05 | **65.92** |
| + SER[‡] | 64.00 | 62.89 | 60.60 | 60.01 | **37.02** | 56.65 | **53.40** | 58.81 | 57.43 | 65.06 | 87.36 | 79.14 | **79.60** | 79.82 | **80.42** | 65.48 |

Table 16: Top-5 accuracy comparison on ImageNet-P, including 14 perturbation types, for our method and other equivariant representation learning methods built upon the invariant representation learning baseline MoCo (He et al., 2020).

| Algorithm | Noise | | | Blur | | | Weather | | | Digital | | | | | Avg. |
|---|---|---|---|---|---|---|---|---|---|---|---|---|---|---|---|
| | Gau. N. | Shot | Speck. | Motion | Zoom | Gau. B. | Snow | Spatter | Bright. | Trans. | Rot. | Tilt | Scale | Shear | |
| MoCo-v3 | 87.75 | 87.81 | 87.82 | 80.34 | 87.91 | 87.94 | 79.22 | 86.75 | 84.54 | 87.72 | 85.16 | 87.48 | 69.08 | 85.84 | 84.67 |
| + AugSelf | 87.58 | 87.50 | 87.59 | 80.81 | 87.73 | 87.67 | 79.70 | 86.50 | 83.07 | 87.66 | 85.25 | 87.31 | **71.44** | 85.83 | 84.69 |
| + STL | 86.67 | 86.65 | 86.66 | 78.33 | 86.82 | 86.77 | 77.62 | 85.59 | 83.21 | 86.50 | 83.81 | 86.50 | 66.31 | 84.75 | 83.30 |
| + SER | **88.62** | **88.59** | **88.60** | 82.01 | **88.66** | **88.68** | 80.56 | 87.58 | 85.59 | 88.52 | 86.01 | 88.17 | 71.25 | **86.78** | **85.69** |
| + EquiMod[†] | 88.78 | 88.79 | 88.75 | 80.86 | 88.93 | 88.98 | 80.16 | 87.95 | 83.41 | 88.68 | 86.38 | 88.51 | 71.89 | 87.13 | 85.66 |
| + E-SSL[‡] | 89.60 | 89.58 | 89.64 | **83.94** | 89.78 | 89.72 | 82.95 | 88.80 | 86.88 | 89.47 | 87.25 | 89.31 | 73.39 | 87.79 | 87.01 |
| + SER[‡] | **89.96** | **89.93** | **90.00** | 82.50 | **90.10** | **90.08** | 83.23 | 89.15 | 87.30 | 89.93 | 87.80 | 89.68 | 75.80 | 88.29 | **87.41** |

## A.6 Latent Space Visualization

Beyond quantitative metrics, we provide a qualitative visualization of latent features extracted from MoCo (invariance only) and MoCo+SER. Due to ImageNet's large class count (1000), we randomly sample 20 classes for visualization. As shown in Figures 4 and 5, the resulting embeddings show clearer qualitative separation for several classes under SER, consistent with improved linear evaluation and robustness.

Table 17: **Computation overhead.** Measured FLOPs includes both forward and backward pass with a 2-view augmentation policy, and "Relative overhead" is the relative FLOPs to vanilla MoCo-v3. FLOPs for SER were measured for the overall mini-batch computation and divided by the mini-batch sample number (including both b1 and b2 as illustrated in Figure 2)

| Method | Per-image FLOPs | Relative overhead |
|---|---|---|
| MoCo-v3 | 18.48G | 1.0x |
| + SER | 18.63G | 1.008x |

## B LIMITATIONS

SER relies on structured geometric transformations (rotations, scaling, flips), limiting applicability to modalities or tasks where such transformations are meaningful. Because scaling is implemented via discrete resampling/interpolation on a finite grid, its invertibility holds only up to discretization effects. Extending the approach to domains without clearly defined group actions (e.g., text, audio, graphs) is non-trivial. While SER is designed to be lightweight, it introduces additional computation; in our main setting the measured overhead is small (Table 17), but any added cost can matter in resource-limited regimes.

## C USE OF LARGE LANGUAGE MODELS

We used large language models (LLMs) to provide writing assistance during the preparation of this manuscript. The LLMs were used in the following ways:

- Polishing and rephrasing sentences for clarity and readability, including parts of the introduction, background, and experiments.
- Condensing text to meet page limits.

Importantly, the LLMs were not used for research ideation, experimental design, implementation, or result generation. All conceptual contributions, algorithm development, theoretical analysis, and experimental work were conceived, conducted, and verified entirely by the authors.

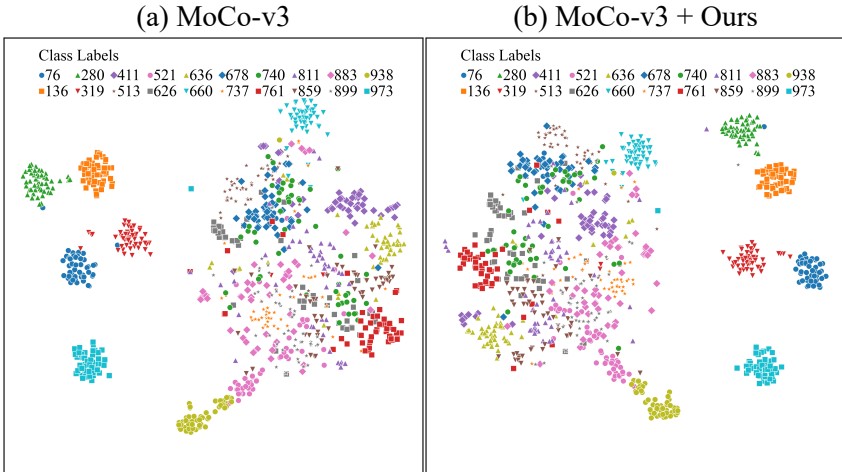

Figure 4: t-SNE visualization of latent space features from 20 randomly sampled ImageNet-1k classes, comparing (a) MoCo-v3 (trained with invariance loss alone) and (b) MoCo-v3 + Ours. Our method promotes better class clustering, demonstrating that incorporating equivariance benefits downstream tasks requiring invariance.

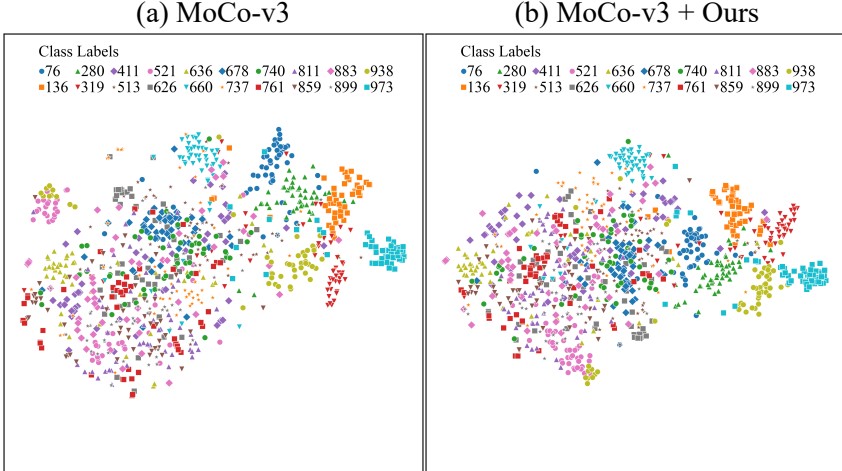

Figure 5: t-SNE visualization of latent space features from 20 randomly sampled ImageNet-C classes under defocus blur corruption, comparing (a) MoCo-v3 (trained with invariance loss alone) and (b) MoCo-v3 + Ours. Our method maintains better class clustering under corruption, demonstrating robustness benefits of incorporating equivariance.

