# OpenReview forum: "Soft Equivariance Regularization for Invariant Self-Supervised Learning"
_ICLR.cc/2026/Conference — ICLR 2026 Poster_

### Official Review · Reviewer_BZu2 · 2025-10-28

**Soundness:** 3
**Presentation:** 3
**Contribution:** 3
**Rating:** 6
**Confidence:** 3

**Summary:**

The authors present a simple yet effective method for introducing an equivariant regularisation constraint into self-supervised ViTs. The goal is to better decouple the traditional approach of simultaneous invariant and equivariant learning on the output embedding space by enforcing intermetiate layers to be equivairant to the input transformations by preserving the spatial structure, while the output vector remains invariant. The approach demonstrates strong invariant performance on downstream linear evaluation tasks, and scales to imagenet level datasets. However, its primary weakness is the lack of a direct evaluation to confirm that the intermediate layers actually learn the desired equivariant properties, and its motivation for avoiding transformation labels is partially contradicted by its own methodology.

**Strengths:**

- The analysis and subsequent conclusion that enforcing equivariant at the output embeddings is a significant and interesting contribution to the equivariant self-supervised literature. This provides new insights and research directions that the community will find valuable.
- The method presented is simple yet effective at improving downstream invariant performance. The approach is generalisable, and enforces the desired properties at defined layers within the network. The enforcement of equivariant feature maps at intermediate layers is seemingly correct.
- The empirical results demonstrate the effectiveness of the method and its ability to scale, where prior works have not yet managed to achieve competitive performance on large-scale datasets.
- Limitations are presented and discussed
- Code and implementation details are provided for validation and reproduction.

**Weaknesses:**

**Major:**
- It is mentioned in the introduction one of the drawbacks is that knowing the transformation labels is an inherent drawback, however, the proposed method also requires the transformation group to be known. While not necessarily a weakness in my opinion, this contradicts the authors motivation.
- No evaluation of the equivariant property is performed. For example, how well do the intermediate representations which the regularisation has been applied to and the invariant representations encode the translation information. It cannot be claimed that equivariance has been enforced without some analysis of the inversion property at a minimum.

**Minor:**
- I would argue that the regularisation is not necessarily soft. The authors are specifically enforcing features to preserve the transformation in the latent space of the intermediate layers via a direct distance metric. This seems like a specific hard constraint.
- The additional computational overhead is presented as a limitation but not quantified.
- It is not clear how the 3DIEBench 3D tranformations are encoded into the equivariance regularisation. Are the these object transformations encoded in the g parameter or are the standard crops and rotations applied?
- See questions for additional minor weaknesses.

**Questions:**

1. In practice how do you determine which layers to apply the soft-equivariant objective to? For some transformation groups the layers which have the most significant effect may change, therefore in practice this is a heuristic / empirical pre-processing step to identify? How does the layer performance change under different architectures and datasets?
2. You do not want to operate the equivariant objective on the pooled feature vector of the output. Hence the question is, why not just omit the final pooling/mean layer? Also prior works such as Konstantinou, et al., (2025) EquiCaps: Predictor-Free Pose-Aware Pre-Trained Capsule Networks. perform on the feature embedding prior to pooling hence maintaining the spatial component.
3. How does the model perform if you apply the equivariant regularisation on all layers of the equivariant network? Why perform on just one when you can enforce all features?

---

> ### Author Response · Authors · 2025-11-22
> **Key Contributions**
>
> We appreciate the reviewer for the constructive feedback.
>
> In the recent line of works that aim to *enrich invariant SSL with structured equivariant representations* (EquiMod, SIE, STL, AugSelf, ESSL, etc.), SER contributes a simple and scalable soft equivariance regularization applied via group action using a spatially structured intermediate representation. We (1) highlight the main distinctions of SER here and (2) address the raised questions/weaknesses in separate, follow-up comments:
>
> ---
>
> **[D1] Decoupling equivariance and invariance via intermediate features**
>
> SER encourages equivariance at intermediate representations while keeping the final representation trained purely for invariance with a standard SSL objective. We empirically show that imposing equivariance and invariance on the same final layer is sub‑optimal: moving the equivariance loss toward the output increases equivariance scores but degrades ImageNet Top‑1 accuracy (Fig. 3, Tab. 4). This motivates our design choice of regularizing an intermediate layer instead of the final representation.
>
> **[D2] Group Action Rather Than Transformation Information**
>
> Building on previous works (e.g., AugSelf, STL, EquiMod, ESSL), SER instead operates directly with mathematically straightforward group actions $\rho_g$ on spatially-structured patch tokens. Because we know $\rho_g$ analytically on these tokens, SER does not require learning of transformation labels/codes with an additional module; the encoder is regularized solely via the group action applied to spatially-structured feature maps.
>
> **[D3] Practical significance and robustness gains**
>
> Despite its simplicity, SER yields consistent and practically meaningful gains. It improves ImageNet‑1k linear and nonlinear evaluation, robustness benchmarks (ImageNet‑C/P), COCO detection, and 3DIEBench, while adding negligible compute overhead. In 2‑view setting, SER outperforms both the invariant baseline and prior equivariance methods (Tab. 1) and also improves other invariance SSL algorithms, i.e., DINO and Barlow Twins (Tab. 2). We additionally report 3‑seed statistics on ImageNet‑1k, confirming that these improvements are stable.

---

> ### Author Response · Authors · 2025-11-22
> **Responses to Reviewer BZu2 Comments [1/2]**
>
> **[Major 1] Known group vs “no transformation labels” contradiction**
>
>
> Our intention with the phrase “no explicit transformation labels” follows the terminology of STL [1]. We do not claim that knowing the augmentation operators themselves is a drawback; these are part of the standard SSL pipeline and are known by construction in all methods, including ours. The issue we highlight is *learning or predicting per‑sample transformation labels/codes as targets*.
>
> SER assumes a known geometric group $\mathcal{G}$ derived from the standard augmentation pipeline (as is also done in prior explicit equivariant methods), but it does not introduce any additional transformation‑prediction head or supervision signal. Instead, it directly applies the relative group action $g = g_2 g_1^{-1}$​ to intermediate feature maps and penalizes deviations from the equivariance relation, using only the transformations already sampled by the SSL framework (zero cost). We will clarify this in the introduction by replacing “requires transformation labels” with “requires learning/predicting transformation labels” and by explicitly stating that SER assumes a known geometric group $\mathcal{G}$ but avoids learning or predicting transformation labels.
>
> **[Major 2] No evaluation of equivariant property**
>
> Thank you for pointing this out. We do in fact explicitly evaluate how well the learned representations satisfy the equivariance relation, although this may not have been sufficiently emphasized.
> * Appendix B.2 defines a group-theoretic equivariance score directly measuring $f(\rho_g(x)) \approx \rho_g(f(x))$ with $g = g_2 g_1^{-1}$​. Concretely, we measure $\boldsymbol{E}_{x, g} \big[d(\rho_g(f(x)), f(\rho_g(x)))\big]$, $  g = g_2 g_1^{-1}$ using cosine similarity for the group elements in $\mathcal{G}$ (rotation $90^{\circ}$, horizontal flip, and scaling).
>
> * Table 4 reports this score on the final [CLS] representation when varying the regularized layer: moving $L_{\text{equiv}}$ closer to the output monotonically improves the equivariance score for all transformations in $\mathcal{G}$, but reduces discriminative quality. This represents 1) the equivariance regularization on intermediate representation propagates to the final representation, and 2) a trade-off between transformation sensitivity and discriminative power.
>
> * Table 5 further probes transformation encoding by training lightweight linear classifiers on frozen layer-wise features to predict rotation/flip labels. Even though $L_{\text{equiv}}$ is applied only at layer 3, intermediate and later layers achieve very high accuracy (>99 % for rotation in many layers), confirming rich structured encoding of the group elements.
>
> * Since all actions in $\mathcal{G}$ are invertible, the inversion property holds exactly by construction. Translation sensitivity of the invariant [CLS] representation is evaluated indirectly via the translation-specific perturbations in ImageNet-P (Tables 8,9), where SER shows clear gains (+1.22 compared to MoCo baseline in average).
>
>
> **[Minor 1] Is this really ‘soft’ regularisation?**
>
>
> Thank you for raising this point. In our terminology, following Finzi et al. [2] and Kim et al. [3], we use “hard” to denote architectural equivariance constraints, such as group CNNs [4] where $f(\rho_g(x))=\rho_g(f(x))$ holds exactly by construction for all $g$. In contrast, “soft equivariance” denotes loss-based regularization that biases the model toward equivariance but allows deviations controlled by a weight $\lambda$.
> SER falls into this latter category: $L_{\text{equiv}}$ is an additive penalty weighted by $\lambda$, and setting $\lambda=0$ exactly recovers the invariant baseline. Moreover, Table 4 shows that the resulting equivariance scores remain clearly below 1.0 at the final layer, confirming that the constraint is not enforced exactly but encourages soft equivariance with a tunable strength. We will clarify this distinction in the revision to avoid any ambiguity.
>
> **[Minor 2] Computational Overhead**
>
> We agree that the overhead should be quantified. MoCo-v3 costs 18.48 GFLOPs/image (forward+backward with 2-view augmentation). Adding SER increases this to 18.63 GFLOPs/image, i.e. +0.8 % overhead (0.15 GFLOPs/image). This matches the expected ~4× the 4.6 GFLOPs single-crop forward cost reported in the MoCo v3 paper. Tab.12 was added in the Appendix with this result.
>
> ---
>
> [1] Yu et al., “Self-supervised Transformation Learning for Equivariant Representations”, 2024
>
> [2] Finzi et al., “Residual Pathway Priors for Soft Equivariance Constraints”. 2021
>
> [3] Kim et al., “Regularizing towards soft equivariance under mixed symmetries”, 2023
>
> [4] Cohen et al.“Group Equivariant Convolutional Networks”. 2016

---

> ### Author Response · Authors · 2025-11-22
> **Responses to Reviewer BZu2 Comments [2/2]**
>
> **[Minor 3] 3DIEBench**
>
> We use 3DIEBench only for downstream linear probing on object class (Table 10) to test out-of-domain transfer. We do not exploit the 3D transformations. Instead, we treat the pre-rendered 2D images with the standard 2D augmentations pipeline on ImageNet. This tests whether the learned 2D equivariance helps even on 3D-rotated objects. Clarification will be added to Sec 4.3.
>
> **[Q1] Layer choice for $L_{\text{equiv}}$**
>
> Layer selection is indeed an empirical design choice. In our main ViT‑S/16 + MoCo‑v3 setting, we swept the application layer of $L_{\text{equiv}}$ and observed that applying it around the third block yields the best trade‑off: Figure 3 shows a peak in ImageNet‑1k Top‑1/Top‑5 accuracy at this layer, and Table 4 shows that moving $L_{\text{equiv}}$ closer to the output further increases the equivariance score but consistently degrades Top‑1 performance.
>
> Across all experiments in the paper we use the same ViT‑S backbone and reuse this single layer choice for different base SSL algorithms (MoCo‑v3, DINO, Barlow Twins); Table 2 shows that SER improves all of them without re‑tuning the layer choice, suggesting that the method is not overly sensitive for this architecture. We will explicitly describe this procedure and emphasize that, for new architectures or very different transformation groups, the layer at which $L_{\text{equiv}}$ is applied should be treated as a small hyperparameter search rather than a hard requirement.
>
> **[Q2] why not apply before pooling?**
>
> In our ViT setup, there is no global pooling layer, but the [CLS] token and projection head effectively form a spatially collapsed invariant representation. We deliberately keep this head unchanged and introduce equivariance only at an earlier stage $f^{(1)}$, before [CLS] insertion, where spatial structure is preserved. Figure 3 and Table 4 show that when the equivariant loss is moved closer to the final representation (i.e., making equivariant and invariant objectives act on the same features), equivariance scores of the final representation increase, but ImageNet‑1k Top‑1 accuracy consistently drops.
>
> Related works such as EquiMod and SIE also operate on post‑pooled features, and we observe similar conflicts there: re-implementing EquiMod with its loss applied to an intermediate (3rd) ViT layer instead of the final (12th) layer improves linear accuracy from 68.95±0.02 to 69.51±0.02 (ImageNet-1k, 3 seeds). This supports our design choice to keep a dedicated invariant head and apply equivariance at a separate intermediate layer rather than replacing the pooling/CLS mechanism altogether.
>
>
> **[Q3] Why not enforce $L_{\text{equiv}}$ on all layers**
>
> We have experimented with applying $L_{\text{equiv}}$ to all transformer blocks of ViT‑S/16 and compared it against our best single‑layer choice (3rd block). Under the same pretraining and evaluation protocol, we obtain:
>
> | Linear Eval (ImageNet-1k, 3 seed)  | Top-1 | Top-5 |
> |---|---|----|
> | MoCo v3 | 68.44 ± 0.07 | 88.02 ± 0.04 |
> | + SER (3rd layer) | 69.28 ± 0.01 | 88.79 ± 0.02 |
> | + SER (all layers)  | 69.21 ± 0.05 | 88.78 ± 0.12 |
>
> Though enforcing SER loss on all layers improves final representation quality, but is suboptimal to the single-layer enforcement and enlarges the computational cost. Intuitively, very early layers may not yet encode semantically meaningful features, while very late layers conflict more strongly with the invariant objective (as also seen in Table 4). We therefore choose a single intermediate layer as a principled compromise between effectiveness and efficiency, and will include this empirical comparison in the Appendix of the revised version.

---

### Official Review · Reviewer_XBxz · 2025-10-29

**Soundness:** 3
**Presentation:** 3
**Contribution:** 1
**Rating:** 2
**Confidence:** 4

**Summary:**

- Introduces SER, which decouples invariance and equivariance by applying an equivariance loss to intermediate, spatially structured features while learning invariance on the final head.
- Demonstrates ViT-S/16 pretraining on ImageNet-1k with linear/non-linear evals and robustness suites (IN-Sketch, IN-V2, IN-R, IN-C/P) and an OOD 3D dataset (3DIEBench). Shows consistent gains over MoCo-v3 and positive deltas when plugged into DINO and Barlow Twins; also improves frozen-backbone object detection on COCO.

**Strengths:**

- The paper is well written and easy to follow
- SER does not require transformation labels or predictors; it leverages known group actions on feature maps with a simple NT-Xent loss, keeping the method lightweight and scalable.
- By excluding crop from set of groups $G$ and keeping it in the invariance path, the method respects group theory while preserving the benefits of strong augmentations for representation quality.
- Improves linear/nonlinear evaluation on ImageNet-1k across MoCo-v3/DINO/Barlow Twins and shows better robustness on ImageNet-C/P and 3DIEBench

**Weaknesses:**

- The paper applies rotations, flips, and anisotropic scaling at intermediate feature maps but does not detail how resampling/interpolation, padding patch-grid alignment are handled. On discrete token lattices, 90° rotations are exact but scales are not. The chosen interpolation kernel can materially affect equivariance error. A detailed description and sensitivity study are missing.
- SER avoids predicting labels, yet it relies on knowing $g_2g_1^{-1}$ from the augmentation pipeline and on having a known group action $\rho_g$ at feature level. This limits applicability when transforms are stochastic/unknown (e.g., random photometric transforms, camera jitter).
- ImageNet Top-1 gains over MoCo-v3 are quite modest, especially in 2-view setting. No confidence intervals or multi-seed statistics are provided, so practical significance remains uncertain.
- By construction, SER cannot handle non-group transforms (e.g., crop, typical color jitter) except via the invariance path, so transformation-aware tasks for photometric changes or crops remain unaddressed.
- The $b_2$ path introduces 90° rotations and removes crop. It is possible that the gains partially stem from the augmentation distribution shift rather than the equivariance loss. One should include the same set of augmentations for MoCo-v3 but without
 the equivariant loss i.e. equivalently run SER with $\lambda=0$. Its important to disentangle the effect of augmentations and the equivariant loss.
- There are related papers that put the equivariant objective “earlier” (pre-projector / on spatial features) and keep the invariant objective “later” (post-projector) [1, 2] . The contributions of this work don't seem to be significantly novel compared to existing literature.

[1] Garrido, Quentin, Laurent Najman, and Yann Lecun. "Self-supervised learning of split invariant equivariant representations." arXiv preprint arXiv:2302.10283 (2023).

[2] Devillers, Alexandre, and Mathieu Lefort. "Equimod: An equivariance module to improve self-supervised learning." arXiv preprint arXiv:2211.01244 (2022).

**Questions:**

See weaknesses above

---

> ### Author Response · Authors · 2025-11-22
> **Key Contributions**
>
> We appreciate the reviewer for the constructive feedback.
>
> In the recent line of works that aim to *enrich invariant SSL with structured equivariant representations* (EquiMod, SIE, STL, AugSelf, ESSL, etc.), SER contributes a simple and scalable soft equivariance regularization applied via group action using a spatially structured intermediate representation. We (1) highlight the main distinctions of SER here and (2) address the raised questions/weaknesses in separate, follow-up comments:
>
> ---
>
> **[D1] Decoupling equivariance and invariance via intermediate features**
>
> SER encourages equivariance at intermediate representations while keeping the final representation trained purely for invariance with a standard SSL objective. We empirically show that imposing equivariance and invariance on the same final layer is sub‑optimal: moving the equivariance loss toward the output increases equivariance scores but degrades ImageNet Top‑1 accuracy (Fig. 3, Tab. 4). This motivates our design choice of regularizing an intermediate layer instead of the final representation.
>
> **[D2] Group Action Rather Than Transformation Information**
>
> Building on previous works (e.g., AugSelf, STL, EquiMod, ESSL), SER instead operates directly with mathematically straightforward group actions $\rho_g$ on spatially-structured patch tokens. Because we know $\rho_g$ analytically on these tokens, SER does not require learning of transformation labels/codes with an additional module; the encoder is regularized solely via the group action applied to spatially-structured feature maps.
>
> **[D3] Practical significance and robustness gains**
>
> Despite its simplicity, SER yields consistent and practically meaningful gains. It improves ImageNet‑1k linear and nonlinear evaluation, robustness benchmarks (ImageNet‑C/P), COCO detection, and 3DIEBench, while adding negligible compute overhead. In 2‑view setting, SER outperforms both the invariant baseline and prior equivariance methods (Tab. 1) and also improves other invariance SSL algorithms, i.e., DINO and Barlow Twins (Tab. 2). We additionally report 3‑seed statistics on ImageNet‑1k, confirming that these improvements are stable.

---

> ### Author Response · Authors · 2025-11-22
> **Responses to Reviewer XBxz Comments [1/3]**
>
> **[W1] Interpolation Kernel and Sensitivity Test**
>
> We agree that the exact resampling procedure can affect equivariance error and thank the reviewer for pointing out that our current description is too terse. In our implementation, the anisotropic scaling used for the group action reuses the same resize operator as the baseline SSL method (MoCo‑v3 / DINO / Barlow Twins), including its default interpolation kernel and padding behavior; we do not tune these separately to keep comparisons fair. We will add these details to Sec. 3.1 / Appendix B.2.
> For ViT, we also restrict the scaling factors so that the rescaled image remains aligned with the patch grid (multiples of the 16‑pixel patch stride), so the transformed feature map is again defined on a regular $H_f \times W_f$ lattice without partial patches. Rotations 90 and horizontal flips act purely by permuting patch indices and thus require no interpolation. A systematic kernel‑sensitivity study is an interesting direction, but within our setting we found that reusing the exact data pipeline of the baseline SSL method while only changing which transformations are used for the equivariant branch yields stable improvements (Tables 1-3, 6-8)
>
> **[W2] Knowing Transformation Label**
>
> Our intention in using the phrase “no explicit transformation labels” follows the terminology of STL [1]. We do not mean that knowing the augmentation operators themselves is undesirable; this is assumed in essentially all vision SSL methods. Rather, we aim to distinguish SER from the methods that must learn or predict per‑sample transformation labels/codes as an additional supervision signal, which creates (i) an interdependency between the representation and a transformation predictor and (ii) limited expressiveness for complex compositions, as discussed in STL [1].
> SER instead assumes the same known augmentation pipeline as the base SSL method and a known geometric group $\mathcal{G}$ acting on feature maps, but *does not learn or predict transformation labels.* For each pair of views, we use the relative geometric transform $g=g_2g_1^{-1}$ induced by the standard **stochastic** geometric augmentations already used in SSL (anisotropic scaling from RandomResizedCrop without cropping, RandomHorizontalFlip, and discrete rotation 90), and directly apply the corresponding action $\rho_g$ to the intermediate feature map. The augmentations are sampled stochastically, but once sampled, the transform $g$ is fully known; no additional transformation‑prediction head is required.
> Stochastic non‑group transforms (e.g., color jitter, blur, grayscale, solarization) are handled exactly as in the underlying SSL method: they do remain in the augmentation pipeline but only contribute to the invariant loss $L_{\text{inv}}$, since no group action $\rho_g$ is defined for them. This matches the scope of this work (and related methods such as AugSelf, STL, EquiMOD, E-SSL), which is to incorporate equivariance for a *known geometric group* on top of an invariance‑based SSL framework, rather than to model arbitrary unknown perturbations such as camera jitter. For invertible/non-invertible transformations, please refer to **[Q3] invertible/non-invertible transformations** of Reviewer `EYXE`.
>
>
> **[W3] No multi-seed statistics**
>
> We appreciate this concern and have run 3‑seed experiments for the main ImageNet‑1k linear evaluation (we will update the manuscript accordingly). For MoCo‑v3 (2‑view) and SER (2‑view, same architecture and training budget) we obtain:
>
> | Linear Eval (3 seed) | MoCo (2-view)  | MoCo+Augself (2-view) | MoCo+STL (2-view) | MoCo+SER (2-view)  | MoCo+Equimod (3-view)  | MoCo+ESSL  (2+4-view)   | MoCo+SER (2+4-view) |
> |-----------|------------------|-------------------|-------------------|-------------------|-------------------|-------------------|-------------------|
> | Top-1   | 68.44 ± 0.07 | 67.55 ± 0.05  | 65.49 ± 0.12 |  69.28 ± 0.01 | 68.95 ± 0.02  | 70.6 ± 0.04 |  71.56 ± 0.03 |
>
> | Linear Eval (3 seed) | DINO (2-view)  | DINO+SER (2-view) | BarlowTwins (2-view) | BarlowTwins+SER (2-view)  |
> |-----------|------------------|-------------------|-------------------|-------------------|
> | Top-1   | 67.37 ± 0.02 | 67.63 ± 0.01  | 63.34 ± 0.03 |  64.02 ± 0.03 |
>
> Considering the standard deviation, the performance improvements are statistically robust.
>
> ---
>
> [1] Yu et al., “Self-supervised Transformation Learning for Equivariant Representations”, 2024

---

> ### Author Response · Authors · 2025-11-22
> **Responses to Reviewer XBxz Comments [2/3]**
>
> **[W4] Non-Invertible Transformation Sensitivity Task**
>
> We agree that SER, by construction, only models equivariance for transformations that form a group with a well‑defined inverse on the image grid (anisotropic scaling, horizontal flip, rotation‑90°). Non‑group operations such as random crop, standard color jitter, blur, or solarization are therefore handled purely through the invariance path $L_{\text{inv}}$, not through $L_{\text{equiv}}$. In particular, crop changes the spatial support so the relative transform $g=g_2g_1^{-1}$ nd the corresponding $\rho_g$ on the feature map are not well defined.
> Our focus in this work is to study the effect of introducing soft equivariance to a subset of augmentations within standard invariance‑based SSL methods, and to evaluate the impact primarily on classification and robustness benchmarks. Photometric robustness is still evaluated and improved via ImageNet‑C (noise, blur, and digital corruption tracks such as brightness/contrast/fog), where SER consistently improves over MoCo‑v3. For example, the average ImageNet‑C Top‑1 increases from 39.47% (MoCo‑v3) to 40.58% (MoCo‑v3 + SER, 2‑view) and 42.91% in the 2+4‑view setting. Please refer to Table 7 for more details.
> Explicit equivariance to non‑group transforms is an interesting but different problem; we will clarify this scope limitation in the revised paper.
>
>
> **[W5] Disentangling augmentation policy vs. equivariance loss**
>
> We thank the reviewer for raising the concern that our gains might be due to the changed augmentation policy (adding $90^{\circ}$ rotations and removing crop from the equivariant path) rather than the equivariance loss itself.
>
> (i) Augmentation policies for $b_1$ and $b_2$.
> For the invariant baseline, the first sub‑batch $b_1$ follows the same augmentation policy $\mathcal{T}$ as MoCo‑v3, including random crop and all photometric transforms. The second sub‑batch $b_2$ uses  $\mathcal{G} = \mathcal{T} \setminus \{\text{Random Crop}\} \cup \{\text{Rotation 90}\}$ ; we keep the full photometric pipeline (color jitter, grayscale, blur, solarization) and only (i) remove random crop and (ii) add discrete $90^{\circ}$ rotations. Both $b_1$ and $b_2$ contribute to the invariant loss (Algorithm 1: $L_{\text{inv}} = L_{\text{inv1}} + L_{\text{inv2}}$), so we are not training a weaker or different invariance encoder on $b_2$; $b_2$ simply provides extra views for the same MoCo-style invariant objective.
>
> Random crop is treated differently because it does not define a group action in our sense: cropping changes the visible support of the image and is not invertible, so the relative transform $g=g_2g_1^{-1}$ and the corresponding spatial action $\rho_g$ on feature maps are not well‑defined. As a result, we handle crops only through the usual invariance loss on $b_1$, and we do not attempt to enforce “equivariance to cropping”. Our goal is to enforce equivariance for geometric transformations that form a group (rotation, flip, scaling), not to learn sensitivity to non‑invertible transforms such as random crop.
>
> (ii) Control experiment with identical augmentations but no equivariance loss.
> To isolate the effect of the equivariance loss from the effect of the modified augmentation policy, we train a control model “SER ($\lambda=0$)” where:
>
> * the batch is partitioned exactly as in SER,
> * $b_1$ and $b_2$ use the same augmentation policies as SER,
> * the invariant loss $L_{\text{inv}}$ is computed in exactly the same way as SER,
> * but the equivariance regularizer $L_{\text{equiv}}$ is disabled ($\lambda=0$).
>
> Thus, MoCo‑v3 vs. SER($\lambda=0$) measures only the effect of the changed augmentation/batch scheme, while SER($\lambda>0$) vs. SER($\lambda=0$) isolates the contribution of the equivariance loss under *identical* data partition, augmentation policy, and invariance loss.
>
> Linear evaluation on ImageNet‑1k (ViT‑S/16, 3 seeds) gives:
>
> | Linear eval | Top-1 |
> |---|---|
> | MoCo v3  | 68.44 ± 0.07 |
> | + SER, $\lambda = 0$ | 68.82 ± 0.01 |
> | + SER, $\lambda > 0$ | 69.28 ± 0.01 |
>
> The augmentation/batch change alone (MoCo‑v3 $\rightarrow$ SER($\lambda=0$)) accounts for a modest +0.38 pt, while turning on the equivariance loss under the identical augmentation policy (SER($\lambda=0$)$\rightarrow$SER($\lambda>0$)) brings a comparable additional gain of +0.46 pt. This indicates that (1) the augmentation policy and batch partition contribute, but (2) the equivariance regularizer itself is responsible for a substantial and statistically stable part of the improvement.
>
> Finally, as we highlight elsewhere, the relative gains of SER over MoCo‑v3 are even larger on robustness‑focused benchmarks such as ImageNet‑C/P and COCO detection, where geometric equivariance is expected to be most beneficial. This further supports that the equivariance loss, rather than only the augmentation distribution shift, is an important driver of the observed improvements.

---

> ### Author Response · Authors · 2025-11-22
> **Responses to Reviewer XBxz Comments [3/3]**
>
> **[W6] Algorithmic novelty compared to SIE and EquiMOD**
>
> We respectfully disagree with the reviewer’s opinion on the novelty of our method. Our methodological novelties can be categorized into the following two items, and they both distinguish SER from SIE and EquiMOD:
> * **Direct, exact group action $\rho_g$ on spatial tokens (no learned module)**
> Both SIE and EquiMod require training an extra parametric module (hypernetwork+predictor in SIE, FiLM MLP in EquiMod) to implicitly encode the transformation effect from view pairs. SER eliminates this entirely; SER applies the mathematically straightforward $\rho_g$ (given from the augmentation pipeline) directly on intermediate spatial tokens (with height and width). This removes all prediction instability and extra parameters.
> * **Location of Equivariance Loss Application**
> SER regularizes the network by imposing the equivariance loss at an intermediate representation, rather than on the final pooled output. Our ablations (Table 4, Figure 3) show that adding equivariance loss exclusively at the final layer is suboptimal for downstream performance. Both SIE and EquiMOD restrict their equivariant losses to the final output (post-pooling), missing finer-grained spatial regularization. Applying the equivariant loss in SER at intermediate layers yields superior scalability and generalization.
>
> To verify that this design principle is not specific to our own loss, we also reimplemented EquiMod with its equivariant loss moved from the final (12th) layer to our intermediate (3rd) layer, keeping the rest of the method unchanged. In ImageNet-1k with 3-seed and 3-view setting, we found that “equivariance-at-intermediate-layer” insight introduced by SER is useful even for prior methods.
>
> | Linear Eval | Top-1 | Top-5 |
> |---|---|---|
> | EquiMOD, 12th layer  | 68.95 ± 0.02  | 88.87 ± 0.01 |
> | EquiMOD, 3rd layer | 69.51 ± 0.02  | 89.04 ± 0.01 |
>
> Taken together, SER (i) removes the need for learning additional transformation modules by acting directly with group actions on spatial tokens, and (ii) introduces and empirically validates a new design principle: decoupling invariance and equivariance across different layers of a ViT backbone, which is not present in SIE or EquiMod.

---

> > ### Comment · Reviewer_XBxz · 2025-11-26
> >
> > Thanks for the detailed rebuttal. Additions, including 3-seed ImageNet-1k results and λ = 0, significantly strengthen the paper's narrative. However, I am still not fully convinced on two fronts:
> > 1. SER essentially enforces equivariance by applying the same geometric transform ($g_2g_1^{-1}$) directly on the intermediate layer. This hardcodes the feature-space action rather than learning a representation of the group. As a result, SER only supports a small, hand-specified set of geometric group elements (and sidesteps crop/most photometric transforms), which feels quite limiting compared to methods that learn how features should transform.
> > 2. The multi-seed results show the gains are statistically significant, but they still remain quite modest.
> >
> > Overall, I appreciate the clarifications and believe the paper is now clearer and more carefully evaluated. However, due to the restricted form of equivariance and the relatively incremental gains over closely related methods, I retain my score.

---

> ### Author Response · Authors · 2025-11-28
>
> **On hardcoded $\rho_g$ vs. learning a group representation**
>
> Leveraging an analytically known group action $\rho_g$ is a deliberate design choice, in line with many equivariant architectures (e.g., group CNNs [1], SE(3)-Transformers [2]), which encode the transformation law at the architecture level rather than learning it. We adopt the same philosophy at the loss level to softly impose an equivariant inductive bias, with the following benefits:
>
> * Group-theoretic guarantees: Using the analytically known $\rho_g$ guarantees that feature transformations respect the underlying group structure (closure, identity, inverse, associativity), whereas learned transformation networks in place of $\rho_g$ are not guaranteed to satisfy these properties.
>
> * Comparable breadth of equivariant transformations: SER is not more restrictive than prior equivariance‑regularization methods: it supports **three** geometric transforms (horizontal flip,  rotation90, and anisotropic scaling), compared to **one** in ESSL (rotation90), **two** in STL and AugSelf, and **five** in EquiMod. Nevertheless, our focus is on a clean geometric group with well‑defined actions on the feature grid rather than on expanding to arbitrary non‑group photometric operations.
>
> Note that, like STL, SER avoids supervision from explicit transformation labels by leveraging relative transformations. However, STL implements relative‑transformation learning via aligned‑transform batches, where image pairs share the same transformation; an N-image batch therefore yields roughly N/2 distinct relative transformations. In contrast, SER directly operates on $\rho_g$​ using independently sampled $g_1$ and $g_2$​ for each image​, preserving full per‑batch transformation diversity.
>
> ---
>
> **Statistically significant but modest gains**
>
> The statistical significance of the ImageNet‑1k gains is now confirmed (stddev $\approx \pm$ 0.01). More importantly, the improvements are larger on benchmarks that rely more heavily on spatial understanding (e.g., COCO detection and ImageNet‑C/P): +1.7 mAP on COCO detection with a frozen backbone (Tab. 6), +1.22 points on ImageNet‑P (including translation, rotation, scaling; Tab. 8), and +1.11 points on ImageNet‑C (including zoom blur, motion blur, elastic transform; Tab. 7). This matches the goal of SER: to enrich invariance‑based SSL with controlled spatial understanding, rather than to fundamentally change the overall pretraining recipe. Within a fixed 2‑view MoCo‑v3 setting, other equivariant add‑ons actually lower ImageNet‑1k Top‑1 (AugSelf: −0.89, STL: −2.95), whereas SER yields the strongest positive delta (+0.84). Note that ImageNet‑1k gains for equivariant regularizers are typically small; for example, ESSL and EquiMod report average Top‑1 changes of −0.04 and +0.93 over their baselines (with baselines evaluated using fewer views).We will highlight our relative and robustness‑oriented gains more clearly in the revised manuscript.
>
> ---
>
> [1] Cohen et al., “Group Equivariant Convolutional Network”, 2016
>
> [2] Fuchs et al., “SE(3)-Transformers: 3D Roto-Translation Equivariant Attention Networks”, 2020
>
> [3] Yu et al., “Self-supervised Transformation Learning for Equivariant Representations”, 2024

---

### Official Review · Reviewer_t2Vy · 2025-11-01

**Soundness:** 2
**Presentation:** 1
**Contribution:** 2
**Rating:** 4
**Confidence:** 4

**Summary:**

The authors tackle two aspects of equivariant representation learning. First, is scaling to imagenet pretraining, where most works tend to focus on smaller datasets. The second is to not enforce equivariance at a representational level, but partially through the network. The rest of the network is then trained with the final goal being the usual invariance, leading to invariant representations at the end of the network. To enforce equivariance partially through the network, the authors leverage the spatial structure that is kept to apply the same transformation as done on the input.

**Strengths:**

- The proposed approach can be integrated with off the shelf SSL methods and architectures, making it widely usable.

- The goal of learning invariant representations, but still enforcing intermediate representations to be equivariant is an interesting take on equivariant SSL in general.

- The scale of the experiments is appreciated, making it closer to state of the art setups.

- The authors demonstrate performance gains over baseline methods and other equivariant methods, with the caveat that here the representations are ultimately invariant.

- Performance gains are observed over the baseline methods across datasets, with some caveats (see weaknesses)

**Weaknesses:**

1) Line 94: “Our approach requires no explicit transformation labels”. If my understanding is correct, having access to the exact transformation (generated from the labels) is required, which while a different constraint can be even harder to obtain in practice. Perhaps [1] (cited in the paper) should be discussed more, as the authors use a similar idea by applying the transformation to few dimensions of the representations, as defined by the dataset/transformation.

2) Claims of being the first to scale equivariant SSL to ImageNet (abstract, lines 61-63) are incorrect (and unnecessary) as this was already done in [2], training ViT-B models on ImageNet and studying the impact of equivariance on downstream tasks.

3) Unclear loss definitions. $L_{inv}$ is never properly defined, and the choice of negative samples for both  $L_{inv}$ and $L_{equi}$ is unclear. Lines 295-296 indicate that negative pairs are samples from $z$, but then line 300 indicates that “we omit negative pairs sample from the same image as the anchor”. I assume that the latter is correct, but the notation must be made clearer. For $L_{equi}$, how are the negative pairs chosen ? Across the same image or across the batch ? While appendix B.4 clarifies a bit, it should not take this amount of work to understand the loss, which is the core of the paper.

4) The authors mention that cropping is removed from the equivariant sub-batch as it does not form a group, but other commonly used augmentations do not either. The standard protocol of BYOL[3] (used in MoCov3 that is the base of experiments in the paper) includes grayscale, blurring and solarization, all of which definitely do not have an inverse. This would suggest that experiments are performed with suboptimal data augmentations in the invariant batch, which makes the comparison to the baselines more complex/unclear.

[1] Marchetti, Giovanni Luca, et al. "Equivariant representation learning via class-pose decomposition." International Conference on Artificial Intelligence and Statistics. PMLR, 2023.

[2] Garrido, Quentin, et al. "Learning and leveraging world models in visual representation learning." arXiv preprint arXiv:2403.00504 (2024).

[3] Grill, Jean-Bastien, et al. "Bootstrap your own latent-a new approach to self-supervised learning." Advances in neural information processing systems 33 (2020): 21271-21284.

**Questions:**

1) Since $f^{(1)}$ is equivariant, it could be interesting to visualize the feature maps to see how the spatial structure is preserved compared to other approaches/a fully invariant baseline.

---

> ### Author Response · Authors · 2025-11-22
> **Key Contributions**
>
> We appreciate the reviewer for the constructive feedback.
>
> In the recent line of works that aim to *enrich invariant SSL with structured equivariant representations* (EquiMod, SIE, STL, AugSelf, ESSL, etc.), SER contributes a simple and scalable soft equivariance regularization applied via group action using a spatially structured intermediate representation. We (1) highlight the main distinctions of SER here and (2) address the raised questions/weaknesses in separate, follow-up comments:
>
> ---
>
> **[D1] Decoupling equivariance and invariance via intermediate features**
>
> SER encourages equivariance at intermediate representations while keeping the final representation trained purely for invariance with a standard SSL objective. We empirically show that imposing equivariance and invariance on the same final layer is sub‑optimal: moving the equivariance loss toward the output increases equivariance scores but degrades ImageNet Top‑1 accuracy (Fig. 3, Tab. 4). This motivates our design choice of regularizing an intermediate layer instead of the final representation.
>
> **[D2] Group Action Rather Than Transformation Information**
>
> Building on previous works (e.g., AugSelf, STL, EquiMod, ESSL), SER instead operates directly with mathematically straightforward group actions $\rho_g$ on spatially-structured patch tokens. Because we know $\rho_g$ analytically on these tokens, SER does not require learning of transformation labels/codes with an additional module; the encoder is regularized solely via the group action applied to spatially-structured feature maps.
>
> **[D3] Practical significance and robustness gains**
>
> Despite its simplicity, SER yields consistent and practically meaningful gains. It improves ImageNet‑1k linear and nonlinear evaluation, robustness benchmarks (ImageNet‑C/P), COCO detection, and 3DIEBench, while adding negligible compute overhead. In 2‑view setting, SER outperforms both the invariant baseline and prior equivariance methods (Tab. 1) and also improves other invariance SSL algorithms, i.e., DINO and Barlow Twins (Tab. 2). We additionally report 3‑seed statistics on ImageNet‑1k, confirming that these improvements are stable.

---

> ### Author Response · Authors · 2025-11-22
> **Responses to Reviewer t2Vy Comments [1/3]**
>
> **[W1] No explicit transformation labels**
>
> Thank you for this precise comment. Our intention in using the phrase “explicit transformation labels” follows the terminology of STL [1]. We do not mean that knowing the augmentation operators themselves is undesirable; this is naturally assumed in all SSL settings, including ours. Rather, following the terminology of STL [1], we refer to methods *that learn or predict per‑sample transformation labels/codes* as an additional supervision signal, which introduces (i) an interdependency between the representation and the transformation predictor and (ii) limited expressiveness for complex compositions, as discussed in STL [1].
> We will clarify this in the introduction by changing phrases such as “requires transformation labels” to “requires learning/predicting transformation labels,” and by making explicit that SER still assumes a known geometric group $\mathcal{G}$ (as in Sec. 3.1) but avoids learning or predicting transformation labels. We will also expand the discussion of STL in Sec. 2.2, emphasizing that STL learns a dedicated transformation representation, whereas SER directly applies known group actions to intermediate features without an auxiliary transformation‑prediction head.
>
> ---
>
> **[W2] First to scale ImageNet**
>
> We appreciate this comment and agree that Garrido et al. [2] successfully study equivariant world models with ViT‑B encoders pretrained on ImageNet‑1k, and analyze their impact on downstream classification and segmentation performance. Our original wording in the abstract and lines 61–63 was broad. We will add a citation to [2] in the related‑work section and revise the abstract and introduction to remove any “first” claim. Instead, we will emphasize the specific scope of SER: a simple, loss‑based soft equivariance regularizer applied at intermediate ViT layers, which decouples equivariance from the final invariant representation and can be plugged into standard invariance‑based SSL methods (MoCo‑v3, DINO, Barlow Twins) while improving their downstream and robustness performance.
> Note that while Garrido et al. [2] investigate equivariant world models in a JEPA‑style predictive framework, our work focuses on a complementary setting: SER augments existing invariance‑based SSL encoders with an intermediate‑layer equivariance regularizer, without changing their architectures or invariance objectives.
>
> ---
>
> [1] Yu et al., “Self-supervised Transformation Learning for Equivariant Representations”, 2024
>
> [2] Garrido, Quentin, et al. "Learning and leveraging world models in visual representation learning." arXiv preprint arXiv:2403.00504 (2024).

---

> ### Author Response · Authors · 2025-11-22
> **Responses to Reviewer t2Vy Comments [2/3]**
>
> **[W3] Unclear loss definitions**
>
> We thank the reviewer for flagging the ambiguity in our loss notation. We clarify both the invariant and equivariant losses and will revise Sec. 3.3 and App. B.4 accordingly.
> * Invariant loss: The invariance loss $L_{\text{inv}}$ is exactly the standard loss of each base SSL method (MoCo‑v3, DINO, Barlow Twins); we do not modify the definition of positives or negatives for these methods. Our contribution is purely additive: we introduce an auxiliary equivariance regularizer on intermediate features and leave the original invariance objective unchanged.
> * Equivariance loss and negatives: For $L_{\text{equiv}}$, we operate on the intermediate feature maps of the equivariant sub‑batch $b_2$. For each image $i$ and spatial location $j$, we take the patch embeddings $(z_{ij}, z^\prime_{ij})$ from the two views as a positive pair. The negatives for anchor $(i, j)$ are *all* patch embeddings from other images $m \neq i$ in the same sub-batch, across both views and all spatial locations $n$; we omit negatives from the same image as the anchor following [1]. **Please refer to Sec 3.3 with revised equation for $\mathcal{L}_{\mathrm{equiv}}^{i,j}$.** In the revised manuscript, we will (i) replace the previous equation, and (ii) state clearly that negatives for $L_{\text{equiv}}$ are drawn only from other images, while $L_{\text{inv}}$ strictly follows the baseline SSL losses without modification.
>
> **[W4] Unclear definition for $\mathcal{G}$**
> We greatly appreciate the precise and constructive comment. Here, we clarify our policy and address the fairness concern:
> * $b_1$ employs the same augmentation policy $\mathcal{T}$ as the baseline invariance method, e.g., MoCo-v3.
> * The augmentation policy for $b_2$ is $\mathcal{G} = \mathcal{T} \setminus \{\text{Random Crop}\} \cup \{\text{Rotation 90}^\circ\}$ (see Section 3.2 and Figure 2), i.e., full photometric transformations are retained for $\mathcal{G}$. Photometric transformations act only as standard SSL augmentations for which we enforce invariance, as in MoCo-v3; they are not elements of the group on which $\rho_g$ is defined. Note that $b_2$ also contributes to the invariant loss (Algorithm 1: $L_{\text{inv}}$ = $L_{\text{inv1}}$ + $L_{\text{inv2}}$); we are not training a separate or weaker invariant encoder on $b_2$.
> * We define the group action $\rho_g$ only on the geometric subset of $\mathcal{G}$ (i.e., anisotropic scaling, horizontal flip, rotation90), as explicitly listed in the last paragraph of Section 3.1 and reinforced by the Figure 1 caption (“For simplicity, we omit the intensity transformation applied to the original image”).
>
> The last paragraph of Section 3.1 describes only the invertible geometric subset on which the group action $\rho_g$ is actually defined and applied. The current wording (“We set the group $\mathcal{G}$ …”) can be misread as claiming that the full policy is a group.
>
> We will simply rephrase the first sentence of the last paragraph in Sec 3.1 as:
> “We define the group action $\rho_g$ on the invertible geometric subset of $\mathcal{G}$ (anisotropic scaling from ResizedCrop, HorizontalFlip, and Rotation 90); photometric augmentations are retained in the policy $\mathcal{G}$ for $b_2$ but have no associated $\rho_g$.”
>
> Random crop is treated differently because, unlike photometric transformations, it changes the spatial support of the image; as a result, the relative transform $g = g_2 g_1^{-1}$ and the corresponding $\rho_g$ on the feature map are not well-defined. Therefore, we use the crop only for $b_1$ (the standard invariant branch) and remove it from $b_2$.
>
> To directly disentangle the effect of augmentations from the effect of the equivariance loss, we compare SER with a variant where the equivariance loss is disabled ($\lambda = 0$) with baseline MoCo as a reference. These two models share identical batch partitioning, augmentation policies, invariant loss computation; the only difference is whether $L_{\text{equiv}}$ is applied. Over 3 seeds on ImageNet-1k, we obtain:
>
> | Linear eval | Top-1 |
> |---|---|
> | MoCo v3  | 68.44 ± 0.07 |
> | + SER, $\lambda = 0$ | 68.82 ± 0.01 |
> | + SER, $\lambda > 0$ | 69.28 ± 0.01 |
>
> We further note that the relative gains of SER over MoCo-v3 are larger on robustness benchmarks such as ImageNet-C/P and COCO detection (Tables 6-8), where equivariance is expected to be most beneficial.
>
> ---
>
> [1] Pineiro et al., “Unsupervised learning of dense visual representations.” 2024

---

> ### Author Response · Authors · 2025-11-22
> **Responses to Reviewer t2Vy Comments [3/3]**
>
> **[Q1] Visualizing feature maps**
>
> Thank you for this helpful suggestion. In the current manuscript we already evaluate equivariance quantitatively via (i) a group‑theoretic equivariance score based on $f(\rho_g(x)) \approx \rho_g(f(x))$ ​(App. B.2, Table 4) and (ii) layer‑wise transformation prediction accuracy (Table 5). We agree that qualitative feature‑map visualizations would be a valuable complement. In the revised version, we will add an appendix figure that visualizes intermediate feature maps before and after applying $g \in \mathcal{G}$ for both SER and the MoCo‑v3 baseline, to illustrate how spatial structure is preserved under rotations or flips.

---

> > ### Comment · Reviewer_t2Vy · 2025-11-28
> >
> > Thank you for your answers, that have helped improve the presentation of the method and clarify the experimental setups.
> > The additional results with $\lambda = 0$ are particularly useful to make the performance gains clearer (as we already see a gain over the baseline), and i would encourage the authors to clearly incorporate these results in a revised manuscript.
> >
> > We are currently unable to update our scores, but I will raise my score to 6 in light of the rebuttal.

---

### Official Review · Reviewer_EYXE · 2025-11-01

**Soundness:** 3
**Presentation:** 4
**Contribution:** 3
**Rating:** 8
**Confidence:** 3

**Summary:**

This paper proposes using soft equivariance regularization (SER), where output features use an invariant loss while intermediate features use equivariant regularization in the context of SSL. For transformers, the output features are spatially collapsed and thus in order to use both invariance and equivariance objectives commonly used in SSL, the authors encourage equivariance only in the intermediate representations, which are then passed to an invariant block that also takes [CLS] token as input. The authors show results on ViT-S on ImageNet-1k with linear and nonlinear evaluations, and compare against various SSL baselines.

**Strengths:**

- Practical way to use transformers for invariance and equivariance objectives in SSL, considers spatial collapse and distinguishes between invertible and noninvertible augmentations.
- Experiments are done at ImageNet scale.
- The evaluation is pretty comprehensive: considers linear and nonlinear evals, robustness (ImageNet-P), and also detection transfer. They consider all the SOTA SSL baselines I'm aware of and also do extensive ablations on the intermediate layer position.
- Does not require transformation labels.

**Weaknesses:**

See questions.

**Questions:**

- The method seems to hinge heavily on $\lambda$ to balance $L_{inv}$ and $L_{equiv}$ and also the batch partition ratio. Is the method sensitive to these parameters?
- It's not clear if regularizing intermediate representations do in fact encourage soft equivariance on the output representations. Is there any way to verify this on the output representations?
- I feel that many SSL methods heavily use non-invertible transformations (e.g. crop, color jitter, random grayscale, mixup/cutmix, etc). Partitioning the batch is reasonable, but does the method rely on using a balance of invertible and noninvertible transformations?
- How does SER fit in with MAE style pretraining? Would you say SER is an alternative to masking for learning representations?

---

> ### Author Response · Authors · 2025-11-22
> **Key Contributions**
>
> We appreciate the reviewer for the constructive feedback.
>
> In the recent line of works that aim to *enrich invariant SSL with structured equivariant representations* (EquiMod, SIE, STL, AugSelf, ESSL, etc.), SER contributes a simple and scalable soft equivariance regularization applied via group action using a spatially structured intermediate representation. We (1) highlight the main distinctions of SER here and (2) address the raised questions/weaknesses in separate, follow-up comments:
>
> ---
>
> **[D1] Decoupling equivariance and invariance via intermediate features**
>
> SER encourages equivariance at intermediate representations while keeping the final representation trained purely for invariance with a standard SSL objective. We empirically show that imposing equivariance and invariance on the same final layer is sub‑optimal: moving the equivariance loss toward the output increases equivariance scores but degrades ImageNet Top‑1 accuracy (Fig. 3, Tab. 4). This motivates our design choice of regularizing an intermediate layer instead of the final representation.
>
> **[D2] Group Action Rather Than Transformation Information**
>
> Building on previous works (e.g., AugSelf, STL, EquiMod, ESSL), SER instead operates directly with mathematically straightforward group actions $\rho_g$ on spatially-structured patch tokens. Because we know $\rho_g$ analytically on these tokens, SER does not require learning of transformation labels/codes with an additional module; the encoder is regularized solely via the group action applied to spatially-structured feature maps.
>
> **[D3] Practical significance and robustness gains**
>
> Despite its simplicity, SER yields consistent and practically meaningful gains. It improves ImageNet‑1k linear and nonlinear evaluation, robustness benchmarks (ImageNet‑C/P), COCO detection, and 3DIEBench, while adding negligible compute overhead. In 2‑view setting, SER outperforms both the invariant baseline and prior equivariance methods (Tab. 1) and also improves other invariance SSL algorithms, i.e., DINO and Barlow Twins (Tab. 2). We additionally report 3‑seed statistics on ImageNet‑1k, confirming that these improvements are stable.

---

> ### Author Response · Authors · 2025-11-22
> **Responses to Reviewer EYXE’s Comments [1/2]**
>
> **[Q1] Batch partition ratio**
>
> In our experiments, SER is not highly sensitive to $\lambda$ or the batch partition ratio $r$ in a reasonable neighbourhood of the default values.
>
> For $r$, we choose a small value (on the order of $10^{-2}$) because each sample in $b_1$ contributes a single [CLS] token, whereas each sample in $b_2$ contributes all 196 spatial tokens (for 224×224), so even a small $\left| b_{2} \right|$ yields many equivariant pairs. Note that larger values of $\lambda$ tend to increase the GPU memory footprint substantially, as the contrastive loss over many tokens becomes memory expensive, which serves as a practical constraint on setting $r$. We will soon update an experimental result for sweeping $r$ as well as for $\lambda$. Note that setting $\lambda=0$ (no equivariance regularizer) only reduces ImageNet Top‑1 from 69.28±0.01 to 68.82±0.01, compared to 68.44±0.07 for the MoCo‑v3 baseline without SER, indicating that the method does not hinge on a finely tuned $\lambda$.
>
> ---
>
> **[Q2] Effect on the equivariance of the output representation**
>
> We explicitly measure soft equivariance at the final representation. Appendix B.2 defines an equivariance score based on the group-theoretic relation $f(\rho_g(x)) \approx \rho_g(f(x))$ with $g = g_2 g_1^{-1}$, and evaluates $\boldsymbol{E}_{x, g} \big[d(\rho_g(f(x)), f(\rho_g(x)))\big]$ using cosine similarity for $d(\cdot)$ with the group elements in $\mathcal{G}$ (rotation 90, horizontal flip, and scaling) at the last layer of the encoder.
>
> As a result, Table 4 reports this score at the final layer when we vary the depth at which $L_{\text{equiv}}$ is applied. As we move the equivariance regularizer from layer 3 toward the final layer, the equivariance score of the final representation increases for all three transformations (rotation, flip, and scale, with scale saturating after layer 9), while the ImageNet‑1k Top‑1 steadily decreases from 69.28 ± 0.01 to 68.18 ± 0.01. This shows that regularizing intermediate features indeed propagates soft equivariance to the output, but there is a trade‑off between equivariance strength and discriminative accuracy.
>
> Additionally, Table 5 trains lightweight classifiers on features from different layers to predict the applied rotation and horizontal flip. Even though $L_{\text{equiv}}$ is imposed only at layer 3, deeper layers (including the final layer) still support very high transformation prediction accuracy (often >99% for rotation), indicating that transformation information induced at intermediate layers is retained by the output representation.

---

> ### Author Response · Authors · 2025-11-22
> **Responses to Reviewer EYXE’s Comments [2/2]**
>
> **[Q3] invertible/non-invertible transformations**
>
> SER does not rely on a delicate balance between invertible and non‑invertible transformations. We keep the full augmentation policy of the underlying invariant SSL method and only distinguish which transforms participate in the equivariance regularizer. Concretely:
> * $b_1$ employs the same augmentation policy $\mathcal{T}$ as the baseline invariance method, e.g., MoCo-v3, including non‑invertible transforms such as random crop, color jitter, and grayscale.
> * $b_2$ employs $\mathcal{G} = \mathcal{T} \setminus \{\text{Random Crop}\} \cup \{\text{Rotation 90}^\circ\}$ (Sec 3.2 and Fig 2); we retain all photometric transforms but remove crop (which breaks the group structure) and add 90° rotation as an additional geometric element. Photometric transforms here act only through $L_{\text{inv}}$; they are not elements of the group on which $\rho_g$ is defined. $b_2$ also contributes to the invariant loss (Algorithm 1: $L_{\text{inv}}$ = $L_{\text{inv1}}$ + $L_{\text{inv2}}$), so we are not training a weaker invariant encoder on this subset.
>
> To assess sensitivity to the choice of invertible (group) transformations, we ablate them on ImageNet‑1k (linear eval, 3 seeds):
>
> | Linear eval | Top-1 |
> |---|---|
> | SER  | 69.28 ± 0.01 |
> | SER -Rot90 | 69.17 ± 0.01 |
> | SER -Rot90-hflip | 69.12 ± 0.12 |
> | SER -Rot90-scale | 68.67 ± 0.01 |
>
> Removing individual group elements (Rot90, HFlip, Scale) leads to only modest degradations: performance remains competitive and well above the MoCo‑v3 baseline. Thus, SER does not depend on a finely tuned mix of specific geometric transforms, as long as a reasonable geometric group is available.
>
> Finally, to disentangle augmentation changes from the effect of the equivariance loss, we compare MoCo‑v3, SER with $\lambda=0$ (same batch split and augmentation policy as SER, but no $L_{\text{equiv}}$), and full SER:
> 	​
> | Linear eval (ImageNet-1k, 3 seeds) | Top-1 |
> |---|---|
> | MoCo v3  | 68.44 ± 0.07 |
> | + SER, $\lambda = 0$ | 68.82 ± 0.01 |
> | + SER, $\lambda > 0$ | 69.28 ± 0.01 |
>
> The modified augmentation usage alone (SER, $\lambda = 0$) does not degrade performance relative to MoCo‑v3, and enabling $L_{\text{equiv}}$ yields a further, statistically significant gain. Mixup/cutmix are not used in our main experiments; if present, they would be treated like other non‑group transforms, i.e., kept in $\mathcal{T}$ for invariance but excluded from $\mathcal{G}$ and the equivariance loss.
>
> ---
>
> **[Q4] Application in MAE**
>
> Thank you for the insightful question. In this work, our scope is aligned with prior equivariant SSL methods such as AugSelf, STL, E‑SSL, and EquiMod: we study how to incorporate equivariance into invariance‑based SSL frameworks (MoCo‑v3, DINO, Barlow Twins), where representations are learned via discrimination / invariance objectives rather than reconstruction. MAE‑style pretraining instead relies on reconstructing masked pixels, and is typically treated as a separate family of reconstruction‑based SSL methods. We therefore do not claim that SER is an alternative to masking; rather, we view SER as complementary to MAE‑style approaches. Extending SER to reconstruction‑style frameworks such as MAE is an interesting future research direction but out‑of‑scope direction for this paper.

---

### Meta-Review · Area_Chair_4CfY · 2026-01-08

**Summary:**

This paper proposes a simple yet scalable method named SER that decouples the two objectives: learning invariant representations via standard SSL, while softly regularizing intermediate features with an equivariance loss. The scores are 8, 6, 2 and 6. The strength of this paper is on the flexibility, no need of transformation labels or predictors, good performance and writing. Considering the comments of all reviewers, I tend to accept.

**Reviewer Concerns:**

The author responded in detail to the reviewers' concerns. some of the reviewers participated in the discussions and raised the scores.

**Reviewer Scores:**

The scores are 8, 4, 2 and 6. But Reviewer t2Vy said to raise the score from 4 to 6 https://openreview.net/forum?id=TuQW7VPfXF&noteId=ZeSZoS9vR9 . So the final scores are 8, 6, 2 and 6.

---

### Decision · Program_Chairs · 2026-01-26

Accept (Poster)